# Global mapping and evolution of persistent fronts in Large Marine Ecosystems over the past 40 years

Qinwang Xing [1], Haiqing Yu [1] ✉ & Hui Wang[1,2,3]

Ocean fronts, characterized by narrow zones with sharp changes in water properties, are vital hotspots for ecosystem services and key regulators of regional and global climates. Global change is reshaping the distribution of material and energy in the ocean; however, it remains unclear how fronts have varied in the last few decades. Here, we present a global, fine-scale digital atlas of persistent fronts around Large Marine Ecosystems and demonstrate significant global increases in both their occurrence and intensity. In subtropical regions (around boundary currents and upwelling systems) and polar regions, persistent frontal occurrence and intensity are rapidly increasing, while in tropical regions, they remain stable or slightly decrease. These enhancements may be respectively related to changes in boundary currents, upwelling, and sea ice retreat. This spatially heterogeneous trend holds important implications for the redistribution of front-related ecosystem services and air-sea interactions but has not been captured by representative high-resolution climate projections models or observation-assimilated ocean models.

Significant warming has been observed in most parts of the world's oceans in recent decades, a trend that is expected to persist in the foreseeable future[1]. This warming and its associated consequences are changing global oceanic circulation and dynamic processes, such as the poleward shift of the ocean gyres[2], changed western boundary currents and east boundary upwellings[3–5], weakened Atlantic Meridional Overturning Circulation[6], and increased upper-ocean stratification[7]. These significant large-scale changes suggest a redistribution of material and energy within the ocean, potentially exerting a profound influence on the reorganization of ecosystems and changes in biological resources[8,9]. In addition to these large-scale processes, climate change may have substantial impacts on discrete oceanographic features such as fronts and eddies, primarily by modifying atmospheric forcing and ocean circulation[10]. These oceanographic features represent ubiquitous high-energy dynamic processes with particularly strong physical-biological-biogeochemical interactions[11,12].

The ocean front is a highly regarded oceanographic feature, defined as a narrow, high-gradient zone (in terms of temperature, salinity and other properties) relative to the surrounding waters,

serving as the three-dimensional boundary of different water masses[13]. The formation of fronts is a result of multiple oceanic and atmospheric forcing mechanisms, including bathymetric gradients, tidal mixing, upwelling, river plumes, water mass convergence, and sea ice freeze-thaw[13]. These fronts, driven by strong water transport and enhanced turbulent mixing, exert significant control over local ocean dynamics and air-sea interactions[14,15], and have been considered as ducts responsible for transporting heat, carbon, oxygen, and other climatically important gases into the deep ocean, as well as facilitating heat transfer out of the ocean[16]. Front-induced upwelling processes inject essential nutrients into the euphotic zone, fueling phytoplankton blooms[17]. In cases where a front is sufficiently long-lived and persistent, front-induced convergence effect will concentrate phytoplankton and zooplankton, ultimately forming high-prey feeding grounds and the biodiversity hotspots to attract the aggregation of fish schools and human fishing activities[13,18]. An expanding body of evidence supports that persistent fronts, also called quasi-stationary fronts (defined as a unique frontal aggregation where fronts frequently occur over relatively long periods) significantly impact the stock variations and

[1]Institute of Marine Science and Technology, Shandong University, Qingdao 266237, China. [2]Southern Marine Science and Engineering Guangdong Laboratory (Zhuhai), Zhuhai 519080, China. [3]National Marine Environmental Forecasting Center, 100086 Beijing, China. ✉e-mail: yuhaiqing@sdu.edu.cn

distribution patterns of fishes and protected marine animals[13,19–21], and have thus been recommended as a priority consideration in the design of marine protected areas[22]. Therefore, clarifying their changes is thus crucial to obtaining a comprehensive understanding and making predictions about marine ecosystem in a changing climate.

Although previous studies have provided examples of frontal variations in some regional oceans, such as the meridional shifts of Southern Ocean fronts[23,24], the southwestward displacement of Patagonian shelf break front[25], and the meridional shift of the Oyashio extension front[26], our understanding of how global fronts will change in term of occurrence and intensity under climate change remains limited, particularly for the aforementioned persistent fronts. One of the primary challenges in investigating their evolution lies in the reliable detection and census of persistent fronts. Satellite observations represent the most effective tool for acquiring information on global frontal activities and monitoring their long-term variations. Previous research has provided a preliminary global perspective on fronts using an automatic front detection algorithm and satellite-observed sea surface temperature (SST) images[27,28]. What is most remarkable is that numerous persistent fronts have been found within Large Marine Ecosystems (LMEs), and these fronts have further been manually delineated based on global frontal occurrence probability and named according to their geography[13]. Recently, advancements in front detection algorithms and high-resolution satellite observations have significantly enhanced our knowledge of persistent fronts, including the better detection of coastal fronts and the elimination of repeated detections[29], along with the objective delineation method of persistent fronts[30]. These techniques have led to the identification of quite a few new persistent fronts and have corrected previously coarse hand-draw positions[29,30]. However, although existing research has presented global persistent front fields based on the relatively objective and automatic delineation algorithm[30], a comprehensive global atlas with a specific focus on depicting each individual persistent front in LMEs, such as each separately extracted and named front in ref. 13, has not been systematically censused and updated using these persistent front fields. This absence makes it difficult for us to accurately evaluate the long-term changes of each individual persistent front in a changing climate.

Considering the pivotal role of fronts in ocean dynamics, ecology, and management within LMEs renowned by their abundant biodiversity[31–34], in this study, we employed the recently developed detection algorithms of front and persistent front to process 14,610 high-resolution SST images acquired between 1982 and 2021. We developed a framework to generate a fine-scale global digital atlas with spatial resolutions of ~5 km that provides detailed information on the names, locations, and seasonal patterns of 1198 identified persistent fronts around LMEs. Using this atlas, together with global satellite-observed data, reanalysis data, and eddy-resolving Earth system models, we comprehensively assessed the long-term trends of persistent fronts within LMEs under climate change.

## Results and discussion

### Mapping global persistent fronts around LMEs

Utilizing seasonal frontal occurrence fields calculated as the proportion of days with the presence of SST fronts in the last 40 years and the associated persistent frontal segments determined through an objective delineation method (Supplementary Fig. 1), we established a detection framework for persistent fronts (see Method section) and generated a satellite-based dataset (Supplementary Table 1) of global persistent fronts around LMEs to characterize their spatial and temporal patterns (Fig. 1a). A total of 1198 persistent fronts were identified around LMEs in this study, of which 1069 were fully or partly located within LMEs, comprising 80.4% of all identified persistent frontal pixels. This digital global atlas reveals many persistent fronts that were not captured by previous hand-drawn maps (a total of 320 persistent

fronts) that have been extensively cited and used in previous research[13]. The number of identified persistent fronts has significantly increased in most LMEs compared to that of existing hand-drawn maps, with an average increase of 11 persistent fronts (Supplementary Fig. 2a). In addition to the numerical increase, the objective delineation method and improved front detection algorithm contribute to presenting more accurate locations and delineating finer-scale distributions compared to previously hand-drawn persistent fronts[29,30]. Persistent fronts are prevalent in most LMEs, with the exceptions being Central Arctic LME 64 where fronts cannot be found due to the sea-ice cover all year round (see the location and number of each LME in Supplementary Fig. 3). The Barents Sea LME 20 exhibited the highest number of persistent fronts, exceeding 10,000 pixels (40 fronts). Following closely are the Antarctica LME 61 and Mediterranean Sea LME 26, each with over 5800 and 4700 pixels (44 and 70 fronts), respectively. When considering the coverage area of each LME, the Barents Sea LME 20 exhibited the highest number of frontal pixels per unit area, whereas the Antarctica LME 61 demonstrated a moderate level due to its expansive coverage area (Supplementary Fig. 2b).

Occurrence and intensity defined, respectively, as the frequency of presence and gradient magnitude of persistent fronts (see Method section), are pivotal properties determining their dynamic and ecological effects[19,21,35]. Utilizing the global digital atlas of persistent fronts, we computed monthly LME-integrated occurrence and intensity of identified persistent fronts to illustrate their spatial patterns and variations. The average occurrence and intensity of persistent fronts within global LMEs are 3.46% and 1.58 °C/100 km, respectively. The lowest levels are observed within LMEs around polar regions, with an average occurrence of 1.85% and an intensity of 1.16 °C/100 km (Fig. 1b, Supplementary Fig. 4). The frequent presence of sea ice may weaken the formation and identifiability of persistent fronts[36]. The presence of sea ice prevents the measurement of SST from space, making it difficult to detect potential fronts beneath the sea ice. Persistent fronts exhibit low intensity, reaching only 1.10 °C/100 km within tropical LMEs, potentially attributed to spatially uniform heating from strong shortwave radiation[37]. The highest occurrence and intensity of persistent fronts are observed in subtropical LMEs, exceeding 4% occurrence and 2 °C/100 km intensity, primarily associated with the presence of many famous boundary currents and upwelling systems[28]. We selected typical LMEs around boundary currents and upwelling systems and found that persistent fronts exhibit high occurrence and intensity, with an average occurrence of 4.43% and 4.16%, as well as an intensity of 2.20 °C/100 km and 1.92 °C/100 km, respectively (Fig. 1b, Supplementary Fig. 4). Although their average occurrence is only ~1% higher than the global average, considering the area-average definition (see Method) and LME-integrated calculation for persistent frontal occurrence, this difference is sizable and not negligible.

### Long-term trends of persistent fronts

The occurrence and intensity of identified persistent fronts from 1982 to 2021 within global LMEs reveal a significant increase of 0.08% and 0.09 °C/100 km per decade ($P < 0.05$), accounting for 2.43% and 5.55% of their climatology (Fig. 2). The LME-integrated spatial trends in the occurrence and intensity of persistent fronts over the past 40 years exhibit spatial heterogeneity. In tropical regions, persistent fronts display relatively stable trends in occurrence and intensity trends, with increases of only 0.05% and 0.01 °C/100 km per decade. Notably, certain LMEs in Southeast Asia, such as the Indonesian Sea LME 38, reveal a significant decrease in occurrence and intensity by 2.96% and 2.16% per decade. Within temperate and subpolar LMEs, the occurrence of persistent fronts demonstrates nearly stable or slightly increased long-term changes, which are considerably lower than the global trends. However, their intensity shows a noticeable enhancement trend, similar to the global average levels. Subtropical and polar persistent fronts exhibit the most notably increased trends in occurrence and intensity, surpassing

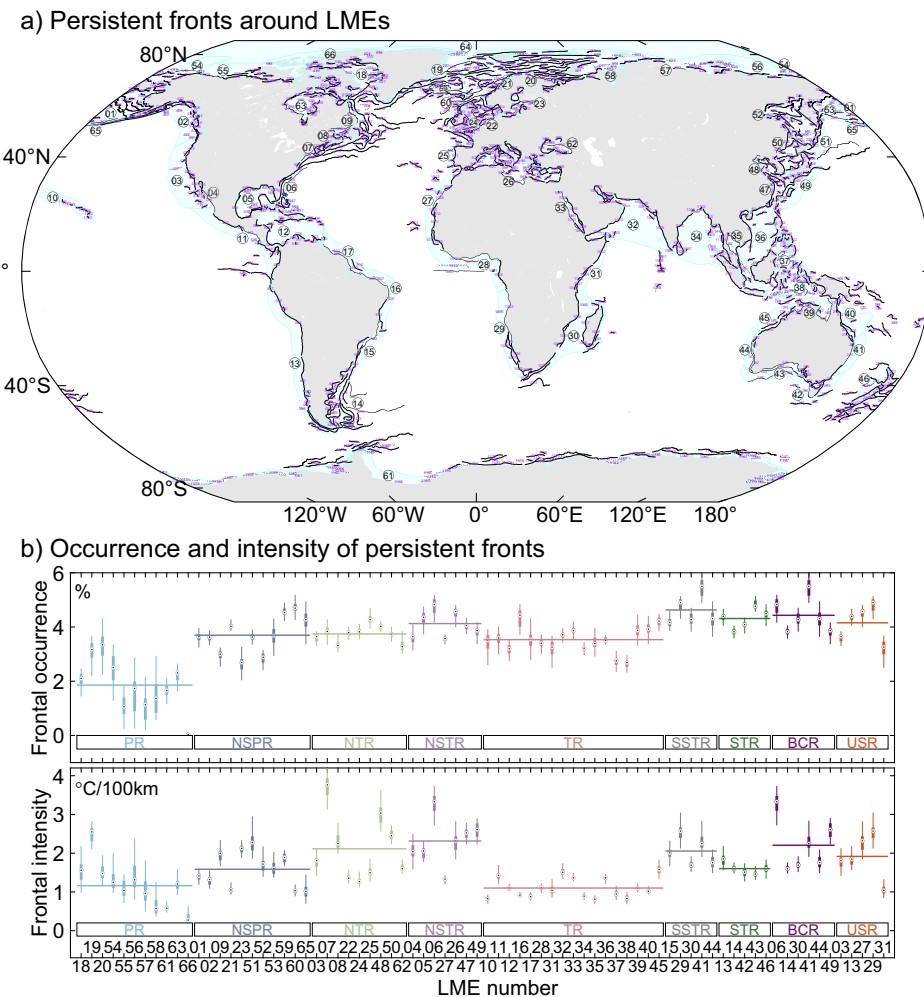

**Fig. 1 | Global persistent fronts around Large Marine Ecosystems (LMEs) identified from frontal occurrence from 1982 to 2021. a** Identified persistent fronts around LMEs. The dashed (solid) lines signify the persistent fronts existing in a (multiple) specific season, and the purple numbers indicate the ID numbers of each persistent front. The detailed names corresponding to each ID number are available in Supplementary Table 1. Blue shadings represent the domain of each LME, while the overlapping circled numbers within each LME denote the LME numbers. **b** Box-plot of annual frontal occurrence and intensity of persistent fronts within each LME.

Each box represents the middle 50% of the data with hollow points denoting the medians, and the whiskers extend to show the maximum and minimum values. The horizontal lines of various colors represent the average results of LMEs in different regions, including polar regions (PR), north subpolar regions (NSPR), north temperate regions (NTR), north subtropical regions (NSTR), tropical regions (TR), south subtropical regions (SSTR), south temperate regions (STR), boundary current regions (BCR), upwelling system regions (USR).

0.11% and 0.12 °C/100 km per decade, significantly higher than the global trends. Within subtropical regions, their occurrence and intensity increase by 3–4% and 5–6% per decade, while in polar regions, these figures surge to 12.28% and 11.58% per decade.

Given that LME-integrated trends can be masked by a limited number of persistent fronts displaying strong long-term signs, we conducted an analysis of long-term variations of individual persistent frontal pixels. The trend distribution within each LME are mainly located in positive values, corroborating the broad enhancement in the occurrence and intensity of subtropical and polar persistent fronts (Supplementary Fig. 5). For frontal occurrence in other regions and frontal intensity in tropical regions, both the proportions of pixels exhibiting positive and negative trends are notably high, suggesting that the long-term changes in persistent fronts exhibit high variability and potential uncertainty within these regions.

**Intensified persistent fronts along boundary currents and upwelling**

The marked alterations observed in persistent fronts within sub-tropical regions primarily stem from LMEs encompassing boundary currents and upwelling systems (Fig. 2). We specifically identified representative LMEs within these regions, revealing a statistically significant upward trend ($P < 0.05$) averaging 3–4% for occurrence and 6–7% for intensity of persistent fronts per decade (Supplementary Fig. 6). Meanwhile, we also observed that more frequent and stronger persistent fronts along boundary currents and upwelling systems correspond to enhanced SST gradients (Supplementary Fig. 7), which may alter the cross-shelf exchange processes and exert a significant impact on the ecosystem in shelf waters[38]. The accelerated strengthening of persistent fronts in these areas is likely linked to the dynamic changes in boundary currents and upwelling systems under global warming[5]. Considering the different dynamic processes between boundary currents and upwelling systems, we further discussed potential dynamic mechanisms in the two types of regions, respectively.

Western boundary currents, such as the Kuroshio current, Gulf Stream, Agulhas current, East Australian current, and Brazil/Malvinas current, play a pivotal role in transporting warm tropical and subtropical waters into the colder subpolar oceans across diverse temperature zones[39]. This process results in the formation of numerous

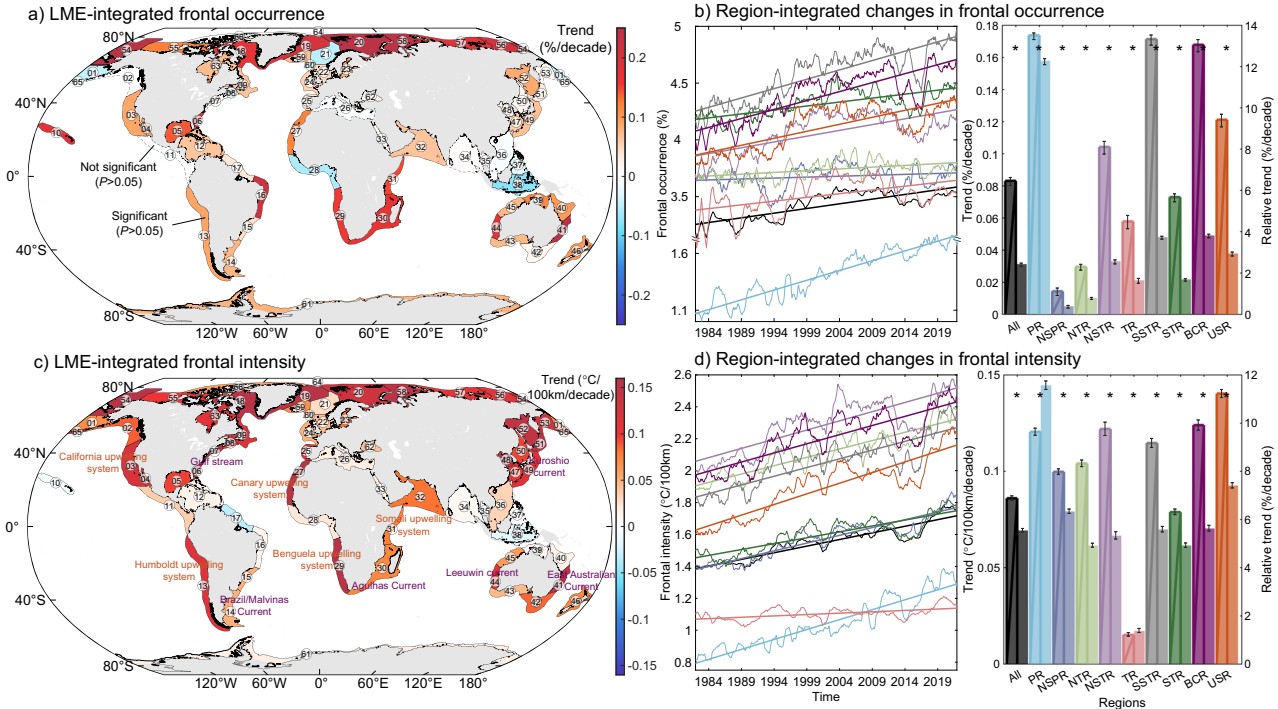

**Fig. 2 | Long-term changes in persistent fronts.** Large Marine Ecosystem (LME)-integrated trends in occurrence (**a**) and intensity (**c**) of persistent fronts. Solid (dotted) boundary lines of the LMEs denote statistically (not) significant trends at the 0.05 level, and the overlapping circled numbers within each LME represent the LME numbers. The low-pass-filtered (>12 month) occurrence (**b**) and intensity (**d**) of persistent fronts and long-term changes. The thick straight lines represent the results of Mann–Kendall test and Sen's slope estimator, while the bar charts with dark borders and without borders represent their linear trend and relative trend, respectively. The black error bars indicate the standard errors of the trends, and asterisks denote statistically significant trends at the 0.05 level. The various colors correspond to different regions, same as Fig. 1b, while the black indicates the average results of global LMEs.

sharp SST fronts around their trajectories[3]. The meander structures of these boundary currents generate a multitude of frontal eddies that, in turn, sustain the presence of oceanic fronts through vertical heat transport[40]. Frequent eddy activity around boundary currents can interact with submarine topography, the boundary currents, and other eddies, often resulting in the formation of fronts[5,40]. The branches from boundary currents can intrude into continental shelf, creating complex frontal systems in adjacent LMEs that separate them from shelf waters[41,42]. Reports indicated that boundary currents have undergone a dramatic warming and acceleration in recent decades, attributed to global warming and increased vertical stratification[3,43]. The intensification of the global wind systems has significantly heightened the eddy kinetic energy of boundary currents, resulting in their broadening[10,39]. In-situ observations have further revealed that a lateral shift of the Gulf Stream towards the shoreward side at a rate of ~5 km per decade[44]. The combined effects of path alterations, broadening of boundary currents, and their rapid acceleration may synergistically contribute to their enhanced intrusion into LMEs situated along the paths of these boundary currents, further leading to the intensification of persistent fronts.

In upwelling systems, the interaction between alongshore wind stresses and the Coriolis force of Earth's rotation propels surface water offshore, allowing cold, nutrient-rich deeper waters to compensate into the upper oceans[45]. The convergence of coastal and open water, as well as the interaction between upwelled and surface water, generates numerous sharp fronts in upwelling systems[31]. Bakun's hypothesis suggests that the faster rise of continental temperatures compared to nearby oceanic temperatures steepens cross-shore pressure gradients, thereby strengthening upwelling-favorable winds under global warming[46]. This hypothesis has been further confirmed to be applicable in a majority of upwelling systems[4]. Despite various evidence indicating considerable uncertainty in the evolution of the main

upwelling systems over recent decades, the widely recognized changes include the enhancement of upwelling in the mid-high latitude regions and weaker or uncertain trends in the low latitude regions of eastern boundary upwelling systems, attributed to the poleward expansion of the Hadley Cells in both hemispheres[47,48]. Additionally, coastal upwelling regions are generally acknowledged as a buffer against global warming that can significantly mitigate the warming trends compared with offshore waters[49]. The contrast in warming trends between upwelling and surrounding regions leads to a stronger cross-shore SST gradient, potentially contributing to the intensification of persistent fronts in upwelling systems. The cooling temperature trends in upwelling regions have also been considered a proxy for enhanced upwelling intensity[47,50]. This study observed significant cooling trends in most coastal upwelling regions located in subtropical LMEs, such as the California upwelling, Humboldt upwelling, Canary upwelling, Benguela upwelling, and seasonal Somalia upwelling (Supplementary Fig. 7), suggesting that the enhanced upwelling may play a key role in intensifying persistent fronts in upwelling systems.

## Rapid enhancing of persistent fronts amidst vanishing Arctic sea ice

Persistent fronts in polar regions exhibit noticeably enhanced trends in both occurrence and intensity, surpassing 10% per decade (Fig. 2). It is important to acknowledge the relatively high uncertainty in satellite-observed SST data in polar regions with high sea ice concentrations[51], where SST is generally more reliable in summer than in winter. To support our findings, we employed the same method on a global daily reanalysis SST dataset covering the period 1993–2021. This reanalysis dataset was derived from an eddy-resolving (1/12°) ocean dynamic model coupling with a sea ice model[52]. In all LMEs, approximately 80% of the persistent frontal pixels identified in satellite-based data are

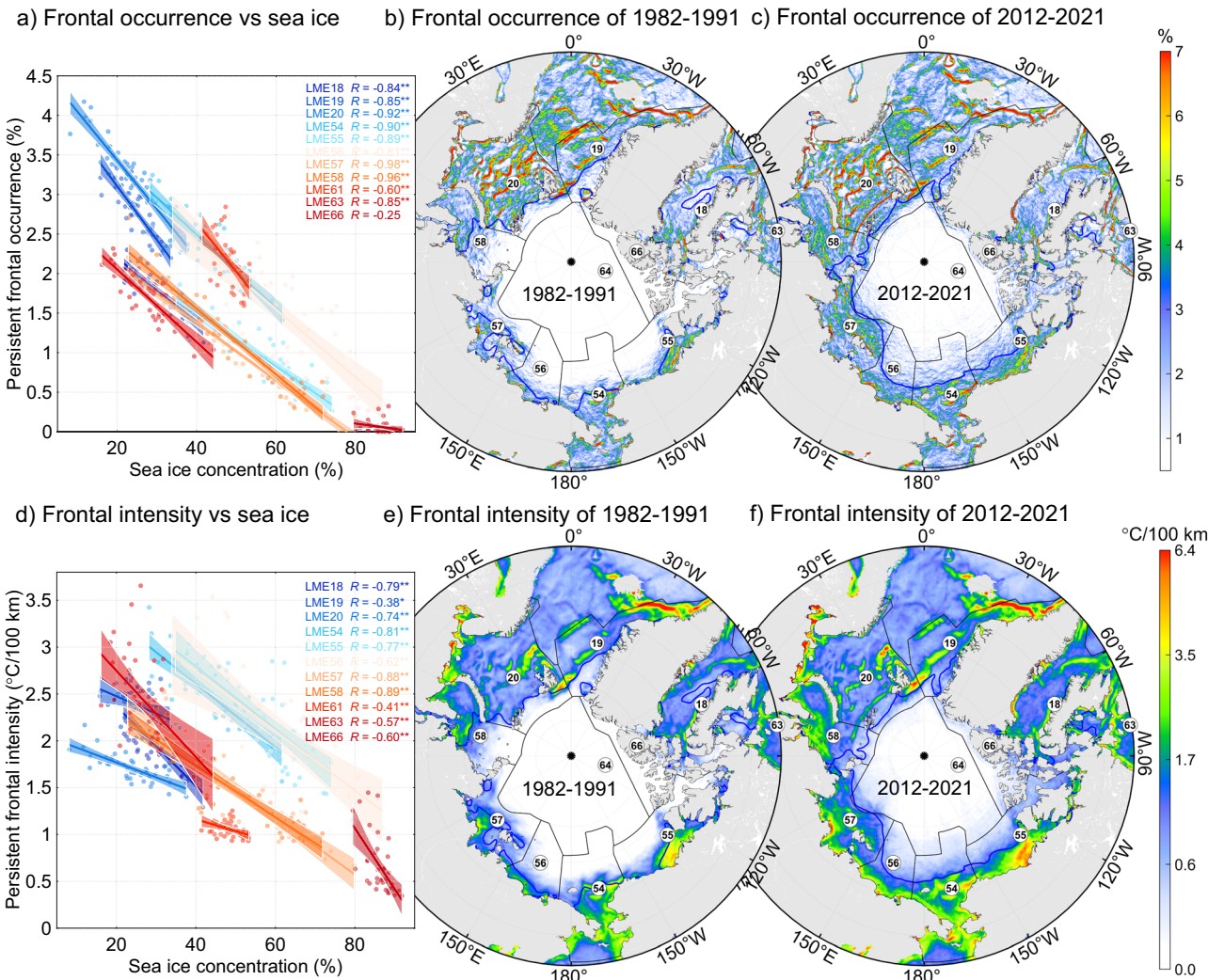

**Fig. 3 | Effects of sea ice on persistent fronts in polar regions. a** Linear regression of annual frontal occurrence vs. sea ice concentration during summer. One and two asterisks indicate that the correlation coefficients are statistically significant at the 0.05 and 0.01 levels, respectively. **b** Spatial distribution of frontal occurrence during summer from 1982 to 1991. Blue lines indicate isogram of 50% sea ice concentration. **c** Same as **b**, but for the period from 2012 to 2021. **d**–**f** Same as **a**–**c**, but for frontal intensity.

located within a 3-pixel proximity to the outcomes of model-based reanalysis data (Supplementary Fig. 8a). This implies that the reanalysis data can effectively capture the distribution pattern of satellite-based persistent fronts within LMEs. The reanalysis SST data also exhibited striking increased trends in the occurrence and intensity of persistent fronts in Arctic LMEs, although persistent fronts in boundary currents and upwelling systems did not show a similar level of enhancement as observed in satellite-based SST data (Supplementary Fig. 9a). The results from this reanalysis data, serving as independent evidence, instill confidence in our identification of persistent fronts and their rapid intensification in Arctic regions.

The Arctic ocean has undergone the most rapid warming among global oceans, occurring at a rate of two to three times faster than the global mean[53]. Arctic sea ice extent and thickness have dramatically decreased at a rate of 12% per decade under global warming[54,55], whereas no significant trend is observed in Antarctic regions[56]. Our correlation analysis indicates that conspicuous intensifications of persistent fronts are closely related ($P < 0.01$) to the reduction of sea ice in all Arctic LMEs during summer (Fig. 3). Comparison of frontal occurrence and gradient fields between the first and last decades over the past 40 years indicates a significant increase in these two properties of persistent fronts in the regions where sea ice concentration

contours have sharply shifted poleward, while this trend is not as pronounced in other regions (Fig. 3). The decline in sea ice and the extension of the ice-free period may enhance the identifiability of persistent fronts that may be covered by sea ice. However, linear trends, even after excluding pixels located in the daily sea ice mask, also show strong increases for persistent fronts in the Arctic ocean (Supplementary Fig. 10), suggesting that the enhanced frontal identifiability resulting from sea ice retreat does not appear to be a major factor responsible for the intensification of persistent fronts. Following global warming and sea ice loss, the circulation dynamics of Arctic ocean are undergoing changes due to increased injection of mechanical energy from the atmosphere and additional available potential energy from temperature and salinity changes. This includes the acceleration of the Beaufort Gyre[57], intensification of near-surface currents[58], and more energetic mesoscale variability[59]. Sea-ice friction can significantly dissipate gravity waves, such as tide, and melting sea have been demonstrated to induce stronger tidal current and tidal mixing, bringing subsurface heat to the surface around Arctic shelf seas[60]. The intensification of these dynamics leads to the strong strain rate, horizontal convergence, or vertical buoyancy mixing, thereby potentially contributing to more available energies for the formation and maintenance of fronts[61,62].

## Future implications

In this study, we developed a satellite-based detection framework to delineate a global, fine-scale digital atlas of SST persistent fronts surrounding LMEs. This openly accessible digital atlas serves as foundational information for advancing research on the effects of persistent fronts on ocean dynamics, marine ecology, biogeochemistry, fisheries, and marine pollution in global or regional waters[14,17,33,61,63]. Additionally, it potentially contributes to the identification of marine conservation areas and marine management strategies[22,33,64]. Utilizing this atlas, we demonstrated that a majority of persistent fronts (853 out of 1198) around LMEs exhibit significant long-term trends ($P < 0.05$) over the past 40 years, while their significantly enhanced hotspots are particularly evident in boundary current, upwelling system, and the Arctic regions. The driving factors of upward or downward trends may vary among different persistent fronts, reflecting their distinct formation mechanisms and changes in local atmospheric and oceanic forces. A thorough exploration into the mechanisms driving the observed evolution of persistent fronts should be conducted in the future, leveraging ample survey data and high-resolution ocean models. In addition to these long-term changes, persistent fronts may also exhibit seasonal and interdecadal variations in response to regional environmental changes and large-scale climate events[31], and their positions may demonstrate spatial shifts under climate change[26]. Our digital map of persistent fronts lays the groundwork for future studies aimed at elucidating the spatial shift, long-term or interdecadal change mechanisms, cross-shelf exchange, and other dynamic or ecological effects of each specific persistent front. It is essential to note that our digital atlas and the observed changes in persistent fronts are based on satellite-observed SST fields due to extended availability and high quality. Consequently, our findings should be confined to SST fronts, while not involving fronts of other water properties or subsurface fronts, despite their usual collocation with different properties (such as physical, chemical, and biological factors) of the same front[13].

Ocean fronts typically instigate intense physical-biological-chemical interactions through strong turbulent mixing both horizontally and vertically[11]. They play a crucial role in providing essential spawning, nursing, and preying habitats for marine animals, forming potential hotspots of marine species and human fishing activities[17]. Our finding of spatially heterogeneous long-term changes in persistent fronts suggests a potential redistribution of ecosystem service provided by fronts, such as fisheries, biodiversity, and carbon sequestration[64]. Ocean warming and consequent deoxygenation compel marine species to expand their distribution ranges poleward[65]. Aside from the impacts of overfishing or inadequate regulations, this climate-driven change in maximum catch potential can have adverse socio-economic consequences, particularly for tropical countries[8]. In comparison to rapidly intensifying persistent fronts in subtropical and polar regions, the decreasing trends for persistent fronts in some tropical regions may exacerbate the vulnerability of countries with limited adaptive capacity, particularly in some LMEs with weakening fronts. Subtropical LMEs located in boundary currents and upwelling systems, being among the most biologically productive marine regions, may magnify their already disproportionately significant contribution to global ecosystem services through the benefits derived from the strengthening of fronts[47]. Ocean warming and sea ice retreat may open up new habitats for sub-Arctic and temperate marine species, while rapidly enhanced persistent fronts act as additional drivers that may promote the climate-induced borealization of Arctic fish communities[9]. It is crucial to further elucidate the role of intensifying persistent fronts in the large-scale reorganization of ecosystems and changes in biological resources induced by climate change. This has become increasingly urgent, particularly given the scenario where persistent fronts in subtropical and Arctic regions may continue to enhance in response to the projected strengthening of upwelling-favorable winds and boundary currents, as well as the projected weakening Arctic sea ice[47,59].

Global change is altering the distribution of material and energy in the ocean[10,43]. The observed trends of persistent fronts offer additional evidence that global ocean dynamics are already undergoing profound changes under climate change. While satellites can observe fronts only at the ocean surface, many frontal structures exist as dynamic processes extending vertically over several hundred meters or even thousands of meters[13]. These features can function as conduits responsible for conveying heat, carbon, oxygen, and other climatically crucial gases into the depths of the ocean, as well as facilitating the transfer of heat out of the ocean[14]. For instance, front-induced subduction and gravitational sinking can greatly enhance carbon sequestration by the ocean[63]. Additionally, they can initiate strong ocean-atmosphere coupled feedback and affect global climate by regulating surface turbulent heat and momentum fluxes[62]. The reinforcement of persistent fronts in subtropical and Arctic regions may have substantial impacts on global atmospheric and oceanic circulation patterns under global warming, especially in boundary currents and upwelling systems that are some of the most significant regions for front-induced air-sea interaction and carbon sequestration[63,66].

However, the satellite-observed trends of persistent fronts cannot be replicated by the eddy-resolving reanalysis product and two high-resolution CMIP6 models used for climate projections (Fig. 4 and Supplementary Fig. 9), which are considered among the most advanced ocean models incorporating observational assimilation and climate models. Specifically, while the observed spatial distribution of persistent fronts can be well simulated (Supplementary Fig. 8), both the reanalysis product and climate models significantly underestimate the intensification of persistent fronts in boundary currents and upwelling systems, only capturing the trends in some polar LMEs. This deficiency in reproducing persistent fronts poses a challenge for climate projections in current Earth system models, potentially introducing biases in long-term trends and internal variability in atmospheric and oceanic circulation, ocean-atmosphere flux, and global climate variability. This limitation may diminish confidence in their application for projected changes in ocean dynamics and marine species under various climate scenarios, and our findings provide a baseline for refining high-resolution Earth system models used for climate projections.

## Methods

### Satellite-observed dataset

Recognizing the potential impacts of variations in cloud contamination and available satellite images on identified fronts[67], we opted to employ a global gap-free SST reprocessed product, made available by the Copernicus Marine Environment Monitoring Service (CMEMS, https://doi.org/10.48670/moi-00169), for the identification of persistent fronts and their evolution. This dataset was developed and processed by ESA SST CCI and C3S, using the Operational SST and Sea Ice Analysis (OSTIA) derived from multiple reprocessed satellite measurements[51,68], including the Sea and Land Surface Temperature Radiometer (SLSTR), the Along-Track Scanning Radiometer (ATSR), and the Advanced Very High Resolution Radiometer (AVHRR). This dataset provides daily average SST at the 20 cm depth with a spatial resolution of 0.05° × 0.05°, spanning Jan 1, 1982 to Dec 31, 2021. It incorporates satellite-based sea ice concentration and mask information from the EUMETSAT Ocean and Sea Ice Satellite Application Facility (OSI-SAF) to mitigate the impacts of sea ice on satellite-observed SST data[69].

### Reanalysis data and climate model data

To support our findings regarding the long-term variations in persistent fronts, we incorporated a daily reanalysis SST dataset from the

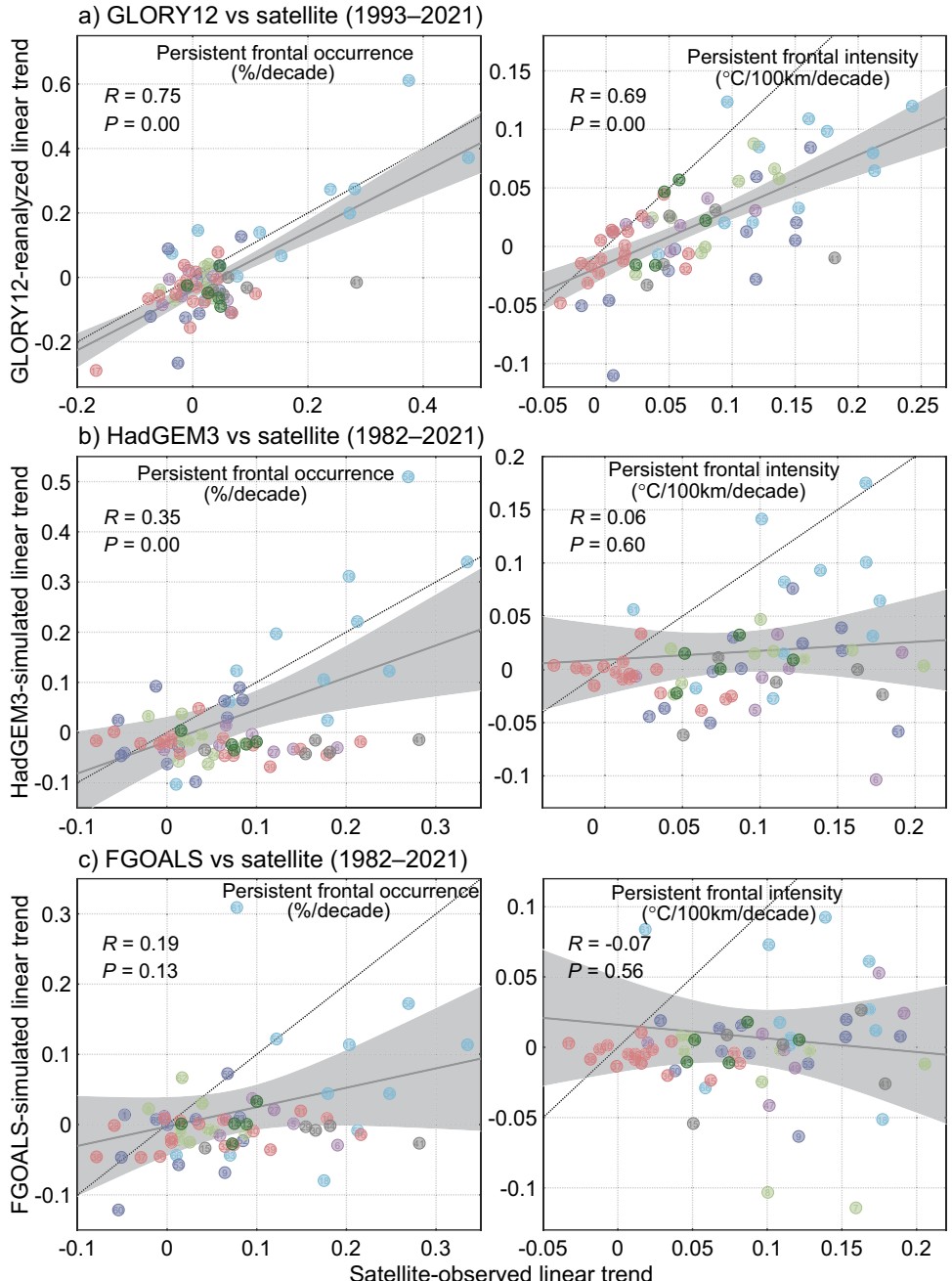

**Fig. 4 | Comparison of persistent frontal changes obtained from satellite observations, reanalysis data, and climate models. a–c** Linear trend comparisons of occurrence and intensity of persistent fronts between satellite-observed data and GLORY12 (**a**) reanalyzed data, as well as HadGEM3 (**b**) and FGOALS (**c**) simulated results in each Large Marine Ecosystems. Thick black lines show linear regression results with 95% confidence intervals in light gray shadings, and dotted lines indicate fitted lines when the two results are nearly equal. The various colors of scatter diagrams correspond to different regions same as Fig. 1b.

Global Ocean Physics Reanalysis (GLORYS12), which was obtained from CMEMS (https://doi.org/10.48670/moi-00021). This reanalysis dataset is based on the eddy resolving (1/12°) Nucleus for European Modeling of the Ocean platform, driven by atmospheric forcing derived from ERA-Interim and ERA5 reanalysis products[52]. It is further constrained through the assimilation of various ocean parameters obtained from satellite and in situ observations, including sea surface height, temperature, salinity, and sea ice data. The time-span of this dataset used in this study spans from Jan 1, 1993 to Dec 31, 2021. This dataset has been proven to be able to effectively simulate interannual and long-term variability in polar sea ice[52]. It should be noted that the potentially time-varying bias in this reanalysis dataset may lead to an

unreliable trend for SST and persistent fronts[70], despite being based on one of the most advanced ocean models incorporating observational assimilation and having been widely applied in previous studies. For example, the rapid increased availability of Argo floats since the 21st century may result in trends that are not due to true changes in the system, and the absence of sea ice thickness in the observational assimilation may make it challenging to accurately depict changes in polar oceans.

Two high-resolution historical simulations from the CMIP6 High-ResMIP experiment were adopted to investigate their performance in simulating persistent fronts. The HadGEM3-GC3.1-HH features a nominal resolution of 8 km with 75 vertical levels and is driven by

atmospheric forcing at a 50 km horizontal resolution[71]. The FGOALS-f3-H boasts a spatial resolution of 10 km with 55 vertical levels and is driven by atmospheric forcing at a 25 km horizontal resolution[72]. We collected daily SST data of HadGEM3 and FGOALS via CMIP6 multi-model database spanning from January 1, 1982, to December 31, 2021 (https://esgf-node.llnl.gov/projects/cmip6/).

## Front detection algorithm

The histogram-based algorithm[73] has been the most popular method over the last two decades and has garnered extensive validation and application in studies related to frontal dynamics and their ecological effects in both global and reginal waters[17,35,74]. Recent enhancements to this algorithm have further refined its capability to identify missing coastal fronts by incorporating inverse distance weighting, while improving frontal continuity and mitigating the issue of repeated detections through the implementation of mathematical morphology operators[29].

This algorithm is designed to identify a boundary that separates two water masses as a front, and consists of five main steps: pre-processing, histogram analysis, cohesion testing, locating frontal pixels, and combining multiple windows[29,73]. In the preprocessing stage, this improved algorithm adopts inverse distance weighting to create buffer zones between SST-available pixels and SST-unavailable pixels (such as land and cloud-contaminated regions) and uses a $3 \times 3$ pixel median filter to reduce the random noise and anomalous values in SST images. Each SST image is then subdivided into numerous $32 \times 32$ pixel windows, ensuring that each pixel is included in five different windows by overlapping 16 pixels. Subsequently, histogram analysis, cohesion testing, and locating frontal pixels are independently conducted in each window to identify potential frontal pixels. The histogram analysis utilizes the ratio of between-cluster variance to within-cluster variance, using a threshold of 0.75 to determine if two distinct SST populations exist in each window. In comparison to a previous study[29], it is worth noting that we introduced an additional treatment where all windows of the two populations with an average SST difference of $<0.25\,°C$ were excluded, aiming to reduce the detection of false fronts in waters with low SST gradients[75]. When both the entire window and the two SST populations exhibit cohesion coefficients exceeding the threshold of 0.92, the window passes the cohesion testing and identifies pixels on the edge of the warm population connected to the cold population as potential frontal pixels. Finally, this improved algorithm consolidates potential frontal pixels from all windows into a binary image and applies a series of morphology operators, including close, thin, and fill operations, to generate the final frontal images.

## The detection framework of persistent fronts

This improved algorithm was applied to daily SST images from 1982 to 2021 for the identification of global fronts. Subsequently, we derived seasonal frontal occurrence fields to further identify global (seasonal) persistent fronts around the LMEs. In this study, Spring, Summer, Autumn, and Winter correspond to the months of March to May, June to August, September to November, and December to February in the Northern Hemisphere, whereas the order is reversed in the Southern Hemisphere. The seasonal frontal occurrence in a specific pixel was established by calculating the total days that the pixel was identified as a frontal pixel within a quarter between 1982 and 2021, and dividing this by the total days in that respective quarter.

A persistent front is defined as a special frontal aggregation where fronts are frequently observed around the same location over relatively long periods[30]. Following this definition, an objective delineation algorithm has been recently developed to identify persistent frontal segments from frontal occurrence images using a combination of the local maximum value method and morphology operations[29,30]. This method employs a $5 \times 5$ pixel Gaussian filter to minimize noise in frontal occurrence and identifies a pixel as a potential persistent frontal pixel if its frontal occurrence value exceeds the average of the neighboring pixels (32 pixels) by 1% in any direction, including the east–west, north–south, southwest–northeast, and northwest–southeast. This threshold value is consistent with previous studies[29,30], and additional comparisons with results obtained using thresholds of 0.7% and 1.3% suggest that the long-term changes in persistent fronts are not sensitive to the selection of the threshold value (Supplementary Fig. 11). Subsequently, morphological close, fill, and thin operations are implemented to generate the final persistent frontal segments with a width of 1 pixel. During this process, the algorithm connects discrete segments separated by gaps of <3 pixels, trims branches with lengths of <5 pixels, and then removes all isolated segments with lengths of <10 pixels.

The above method for detecting frontal segments cannot achieve the comprehensive census of persistent fronts and the creation of a global atlas. Consequently, we developed a framework to merge all the identified persistent frontal segments from the four seasons and compile global persistent front datasets. During this procedure, we merged frontal segments that exhibited variations in location of no more than 10 pixels (~50 km) among the four seasons within local waters into a persistent front, which means that a segment in a season will be identified as an individual persistent front in local waters if its position has a distance of >10 pixels from frontal segments of other seasons. If these frontal segments from the four seasons were in close proximity to each other (<5 pixels) in local waters, we designated the longest frontal segments as the final persistent front; otherwise, if the position of frontal segments varied significantly between seasons (>5 pixels) in local waters, we calculated the average position to determine the final persistent fronts. Additionally, if a branch had a length of >10 pixels, we considered it as a separate, individual persistent front if its average location was at least 10 pixels away from the connected identified persistent fronts. If not, we removed this branch. After completing the identification of persistent fronts, we assigned individual names to each based on their geographic locations and occurrence seasons. Subsequently, we counted the number of occurrence seasons and LME numbers for each front.

We further analyzed the frontal occurrence fields using a reanalysis dataset and two climate model datasets to assess their capabilities in simulating frontal dynamics. This evaluation was based on a comparison of the distribution density between these model-derived fields and satellite-based frontal occurrence fields within LMEs. The hit rate was utilized as a statistical metric to assess the precision of persistent fronts in both reanalysis and climate model datasets within LMEs. It is defined as the percentage of satellite-identified persistent frontal pixels that fall within a 3-pixel radius of those identified in the reanalysis or climate model datasets[28,29].

## Statistical analysis of persistent fronts

Utilizing the daily front data identified in this study, we computed two crucial properties of persistent fronts: frontal occurrence (%) and frontal intensity (°C/100 km). These properties were subsequently analyzed to understand their variations. Frontal occurrence is determined by counting the number of frontal pixels within a 6-pixel radius (~30 km at the equator) of persistent fronts during their respective occurrence seasons and dividing this by the total number of available SST pixels within the same distance during those seasons. Frontal intensity is derived from the SST gradient field in persistent fronts during their occurrence seasons, employing a modified Sobel operator tailored to filter out submesoscale information of <50 km[21,28]. The frontal intensity $\mathbf{MG}_{i,j,n}$ on the $i,j$ location in the input SST image is

intended to eliminate information at spatial scales of $<2n \times \Delta x$, and is formulated as follows:

$$\mathbf{T}_{i,j,n} = \begin{bmatrix} \mathbf{SST}_{i-n,j-n} & \mathbf{SST}_{i-n,j} & \mathbf{SST}_{i-n,j+n} \\ \mathbf{SST}_{i,j-n} & \mathbf{SST}_{i,j} & \mathbf{SST}_{i,j+n} \\ \mathbf{SST}_{i+n,j-n} & \mathbf{SST}_{i+n,j} & \mathbf{SST}_{i+n,j+n} \end{bmatrix} \quad (1)$$

$$\mathbf{G}_{i,j,n,x} = \frac{\begin{bmatrix} -1 & 0 & +1 \\ -2 & 0 & +2 \\ -1 & 0 & +1 \end{bmatrix} \bullet \mathbf{T}_{i,j,n} \times 100}{4 \times 2n \times \Delta x}, \mathbf{G}_{i,j,n,y} = \frac{\begin{bmatrix} +1 & +2 & +1 \\ 0 & 0 & 0 \\ -1 & -2 & -1 \end{bmatrix} \bullet \mathbf{T}_{i,j,n} \times 100}{4 \times 2n \times \Delta x} \quad (2)$$

$$\mathbf{MG}_{i,j,n} = \sqrt{\left(\mathbf{G}_{i,j,n,x}\right)^2 + \left(\mathbf{G}_{i,j,n,y}\right)^2} \quad (3)$$

where, $\Delta x$ represents the spatial resolution (~5 km in this study) of the input SST image, while $\mathbf{G}_{i,j,n,x}$ $\mathbf{G}_{i,j,n,y}$ denote the latitudinal and longitudinal gradient vector components on the $i,j$ location but at spatial scales on the order of $2n \times \Delta x$. The parameter $n$, set to 5 in this study, is designed to filter out information on spatial scales of $<2n \times \Delta x$ (~50 km) to better focus on mesoscale fronts. The inclusion of a factor of 100 in Eq. (2) is used to convert the units of gradients into °C/100 km.

To assess the changes in persistent fronts, we applied a robust nonparametric method, the Mann–Kendall test and Sen's slope estimator, to determine the linear trends and their significance levels for the monthly frontal occurrence and frontal intensity of the persistent fronts[76]. Prior to calculating the linear trends, the monthly frontal occurrence and frontal intensity were smoothed using one-year average filters to eliminate the signal of seasonal variations. We further utilized the linear trend to compute relative trends (%) by dividing Sen's slope by the average frontal occurrence and intensity. Acknowledging potential variations in linear trends along different segments of the same fronts, we individually calculated the trends in frontal occurrence and intensity for each frontal pixel, as well as the integrated trends within each LME, to provide better support for our research findings. Furthermore, all LMEs are categorized into distinct latitudinal regions based on their geographic position to further study the latitudinal variations in the properties of persistent fronts[77]. The latitudinal regions include the polar regions (PR), north subpolar regions (NSPR), north temperate regions (NTR), north subtropical regions (NSTR), tropical regions (TR), south subtropical regions (SSTR), and south temperate regions (STR). The persistent fronts within LMEs around the boundary current regions (BCR) and upwelling system regions (USR) were also categorized into the two groups. It should be noted that, as ecosystem-based management units, the designation of LMEs represents an ongoing scientific process[78], and the number and boundaries of LMEs may evolve over time. For this study, we utilized the currently identified 66 LMEs developed by the NOAA Large Marine Ecosystems Program[79].

Additionally, we conducted Pearson correlation analysis, combined with a significance test, to delve deeper into the potential mechanisms behind the long-term variations in persistent fronts. We examined the yearly sea ice concentration to analyze their impact on the evolution of frontal occurrence and intensity in polar regions. The significant level of the correlation coefficient was adjusted based on their effective degrees of freedom calculated by assessing potential autocorrelation[80], and the effective degrees of freedom formula $N^*_{X,Y}$ between the two time series $X$ and $Y$ is as follows:

$$N^*_{X,Y} = \frac{N}{\sum_{n=-(N-1)}^{N-1} \left(1 - \frac{|n|}{N}\right) \mathbf{R}^X_n \mathbf{R}^Y_n} \quad (4)$$

where, $N$ represents the sample number of time series $X$ and $Y$, while $\mathbf{R}^X_n$ and $\mathbf{R}^Y_n$ denote the $n$-order autocorrelation coefficient of $X$ and $Y$.

## Reporting summary
Further information on research design is available in the Nature Portfolio Reporting Summary linked to this article.

## Data availability
The satellite-observed SST dataset is available from https://doi.org/10.48670/moi-00169, the reanalysis SST dataset is accessible from https://doi.org/10.48670/moi-00021, the Earth system model datasets of HadGEM3 and FGOALS can be accessed from https://esgf-node.llnl.gov/projects/cmip6/. The dataset of identified persistent fronts in this work is available from https://doi.org/10.5281/zenodo.10210578[81], and source data are provided with this paper[81].

## Code availability
The improved front detection algorithm is available from https://doi.org/10.1016/j.rse.2023.113627, the detection algorithm of persistent frontal pixels can be accessed from https://doi.org/10.5281/zenodo.10210578[81], the code of Mann–Kendall test and Sen's slope estimator is accessible from https://www.mathworks.com/matlabcentral/fileexchange/22389-seasonal-kendall-test-with-slope-for-serial-dependent-data.

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

## Acknowledgements

H.Y. and Q.X. was financially supported by the Shandong Provincial Natural Science Foundation (No. ZR2022QD041), the Open Fund Project of Key Laboratory of Marine Environmental Information Technology, and the National Natural Science Foundation of China (No. 42141001). We thank the Copernicus Marine Environment Monitoring Service and World Climate Research Programme for their public datasets.

## Author contributions

Q.X., H.Y. conceived and designed the study; Q.X. compiled all datasets and performed data analysis; Q.X. created and edited main figures and wrote the paper with contribution from H.Y. and H.W.; Q.X., H.Y., and H.W. collaborated in writing and revising the manuscript.

## Competing interests

The authors declare no competing interests.
