## [Peer Review File · Nature Communications]

Global mapping and evolution of persistent fronts in Large Marine Ecosystems over the past 40 yearsREVIEWER COMMENTS

Reviewer #1 (Remarks to the Author):

Review of the manuscript NCOMMS-23-62487-T entitled “Global mapping and evolution of persistent fronts in large marine ecosystems over the past 40 years” by Qinwang Xing, Haiqing Yu, and Hui Wang submitted to Nature Communications

I reviewed two previous papers of the same lead author and made several major comments on methods that have been addressed. Therefore, in this review I’m not going to dwell on methods even though the authors made a few modifications. After all, the authors made their front detection code available, which is commendable. Overall, this work is a major contribution, and it should be published pending minor fixes listed below.

Minor comments:

Line 1-2 and similar instances throughout the paper: “Large Marine Ecosystems” is a standard term, with each word capitalized.

L61: “topographical change” should be “bathymetric gradients”

L62: “melting sea ice” – Consider “sea ice freeze-thaw”

L126: “cover of glacier” should be “sea ice cover”

L131: “Occurrence and intensity, indicating” should be “Occurrence and intensity defined, respectively, as”

L163: “boundary current region” should be “regions”

L209: “around encompassing” should be just “encompassing”

L216: “West boundary currents” should be “Western boundary currents”

L216: “such as the Kuroshio current, Gulf Stream, and Agulhas current” – Consider listing all WBCs.

L342: “contract their suitable habitats and distribution ranges poleward” should be “expand their distribution ranges poleward”

L343: Delete “climate-driven reductions in fish catch” altogether because this statement is false. Many species, especially those that inhabit cold waters, thrive in a warmer climate. For example, the population and catches of some salmonids in the North Pacific exploded lately. Cod and capelin returned to the Barents Sea. Haddock and cod in the NW Atlantic have mostly recovered after the collapse of the

1980s caused in part by extremely cold winters in the early 1980s. Farther south, winter flounder's stock off New England collapsed after two consecutive extremely harsh winters in the mid-1960s; the stock recovered thanks to the subsequent warming. Herring and cod in the Norwegian Sea are doing very well in a warmer climate. All of the above is well documented in the literature. These are just a few examples. Their number can be multiplied. Moreover, fish catches depend on a variety of factors; climate is just one of many factors. Such factors as overfishing or regulations (quotas, restrictions etc.) have a much greater impact on fish catches than climate variability except for such phenomena as El Niño.

L345: "the unclear trends for persistent fronts in tropical regions may exacerbate the vulnerability of countries" – It is unclear to me how "unclear trends" may "exacerbate" anything.

L356: "This has become increasingly urgency..." – urgent not urgency.

L369: "carbon export" should be "carbon sequestration by the ocean"

L562: "boundary current region" must be regions.

L702: "n/a-n/a" – Complete the reference.

===== END of REVIEW =====

Reviewer #2 (Remarks to the Author):

This study presents a global, fine-scale digital atlas of persistent ocean fronts derived from satellite observations and demonstrates statistically significant increase in the occurrence or intensification of persistent ocean fronts, particularly in the subtropics and Arctic regions.

The idea of a fine-scale digital atlas of persistent fronts itself is not actually very unique, as the same approach has been done in Mauzole (2022), who produced a global distribution of persistent fronts using satellite SST fields in association with large marine ecosystem (LME). The central novel aspect of this study, as the authors point out, is the global analysis of long-term trends in the ocean front, which are not very well captured by reanalysis and high-resolution ocean models. Given the crucial role of ocean fronts in ecosystem as well as in the climate system, this study suggests significant implications for various applications.

However, I find that the figures do not efficiently convey the results, and the introduction could benefit from a more refined structure to enhance clarity and engagement. I encourage the authors to make major revisions to the text, especially the introduction part, and figures with a focus on improving clarity and context to better highlight the long-term trend. Some more specific comments are listed below.

Major comments:

1. Introduction

1.1 The context of the introduction is rather discursive, but could be improved by clarifying the points of each paragraph. For example, the first paragraph (especially L49-56) and the second paragraph seem to repeat the physical nature and influence of ocean fronts.

1.2 It is unclear what is meant by a persistent front, although the authors briefly mention it as “sufficiently long-lived and persistent front” in L67-68. Since the thesis of this study emphasizes the trend of persistent fronts as the main finding, an elaboration of the definition of persistent front in plain language (not about the detailed detection algorithm) would help readers to better understand what this study is for.

Regarding the trend, in addition, I also wonder if the trend of frontal occurrence is sensitive to the choice of threshold of “persistence”. Or does this study adopt a widely-used threshold definition or something?

1.3 The authors argue that the global atlas of persistent fronts in LMEs has not been reported based on objective algorithm (L93-95). However, I find that the idea and approach of this study actually follows Mauzole (2022), although the detailed algorithm may differ. Rather, the novelty of this study comes from the verification of the long-term trend of persistent fronts derived from 40 years of global satellite observations including the Arctic region. In this sense, overviews of global changes in LMEs and fronts should be further addressed to strengthen the context regarding the contribution of this study.

2. Clarity of figures

2.1 LME is an important keyword in this study. However, its specific horizontal distribution is shown in Fig. 2, while the regional analysis of fronts in each LMEs is presented in Fig. 1b,c. This figure composition leads to confusion such that the authors explain “LME64” (L125) and “LME 20” (L127) before showing what they are (although the authors refer to Extended Data Fig. 3, it is an inefficient repetition since the same information is presented in Fig. 2). Reorganizing the figures may help to better structure the manuscript.

2.2 Tiny characters and numbers in Figs. 1,2 are hard to discern. I would recommend the authors to consider a more efficient way to leave key information and trim less important things, such as ID numbers of fronts, by moving them to Supplementary figures.

3. Figure 4 presents the results from two models. Although the result of the reanalysis is shown as a horizontal map in the Extended Data figure, expanding Figure 4 to include the reanalysis would clarify the usefulness of the global atlas openly provided in this study.

Minor comments:

1. Reference 3 seems to be incorrect.

2. L118-120: It may be unfair to compare the number of persistent fronts detected in this study with outdated hand-drawn maps, despite the significant improvements in recent studies.

3. L121: Please elucidate what the magnitude of the increase in frontal occurrence as much as 11 indicates (elaborate it in terms of physical meaning, not just adding its unit).

4. Fig 1a caption: meaning of blue shading is missing

5. L167: Please specify a reference of Mann-Kendall test and Sen’s slope estimator. This description of statistics may be moved to Method section.

6. In Fig. 2b,d, please specify the left and right bars in the caption. Also in L202, specify black bars are indicating error bars, because there are another black bars in the figure.
7. L254: Here, do the authors indeed mean “decreasing trend” of upwelling? Or cooling trend? It’s confusing because the next sentence indicates enhanced upwelling.
8. L358: intensity => intensify?
9. I don’t see black lines in Extended Data Fig. 1.

Reviewer #3 (Remarks to the Author):

This paper develops a global dataset of persistent SST fronts in Large Marine Ecosystems (LMEs), typically confined to over and near continental shelves. The authors identify persistent changes in fronts, including changes in location, frequency of occurrence and intensity. Changes are particularly notable in the Arctic ocean, which the authors link to changes in sea ice concentration.

For the most part, this is a fine study with a relatively straightforward analysis. However, I have some concerns about various statements made within the paper for which no evidence is provided. Particularly, the authors attempt to attribute changes in fronts in boundary current regions to changes in upwelling and boundary currents themselves. While both these points are indeed possible, they need to be supported with evidence.

I’ve also noted a few other issues that should be relatively easy to fix or discuss. However, several revisions to the manuscript are required before I could recommend publication.

Major points

Statements without supporting evidence

As mentioned above, there are several statements made by the authors without evidence. I’ve pointed out two in detail below in particular comments, but there may be others. I would encourage the authors to evaluate their manuscript critically, and ensure that all positive statements are supported.

LME Definition

How have the authors defined a Large Marine Ecosystem? There are various, occasionally conflicting, definitions in the literature, and while the authors have cited several references, it is unclear to me where the regions covered by LME have come from. This should be explicit.

Use of Reanalysis Output for analysis of trends

In this paper, the authors use the output of an ocean reanalysis in ice covered regions. While the reanalysis generally supports the satellite observations, it should be noted that reanalysis products are subject to errors in trends due to changes in the observing systems assimilated into them. For example, the authors use reanalysis output from 1993-onwards, but the availability of Argo floats from ~2005 results in trends in the reanalysed output that are not due to “true” changes in the system, but due to Argo coming online and constraining the model. See Wunch

While the authors can't exactly re-run the reanalysis, discussion of these issues should at least be noted and discussed. Trends from reanalysis can be highly unreliable, and sea-ice would be more difficult to interpret still, given the lack of observations of sea-ice thickness.

Particular comments

Line 32: “rapid increases” and “slight decreases” of what? Occurance, intensity, both?

Line 67: I'd probably state “phytoplankton blooms” or just “plankton blooms”, as opposed to “primary production blooms”.

Line 76-77: “our understanding of how they will change under global warming remains limited.” I don't agree with this statement. There is a huge amount of work, particularly in the Southern Ocean, on how fronts will change in the future. See Chapman et al. 2020 for a review, and Graham et al

Line 126: “where fronts cannot be found 126 due to the cover of glacier all year round” do the authors mean sea-ice instead of glaciers?

Line 139: “The frequent presence of sea ice may weaken the formation and identifiability of persistent fronts” Could it also be said that the presence of sea-ice prevents measurement of SST from space? Sea-ice zones, even those are are only seasonally impacted, lose large amount of data simply due to high levels of sea-ice in the winter, and contamination during periods when only low concentrations of sea-ice are present.

Line 145-147: “We selected typical LMEs around boundary currents and upwelling systems and found that persistent fronts exhibit high occurrence and intensity, with an average occurrence of 4.43% and 4.16%” The authors suggest here that an occurrence of 4.4% is “high”. However, that's compared with a global average of 3.4%. If we translate this to number of days with a front per year, that's 16.2 days/year vs a global average 12.6 days/year. While higher than the global average, this doesn't strike me as particularly “high” and probably only borderline statistically significant.

Line 229-233: “The combined effects of path alterations, broadening of boundary currents, and their rapid warming and acceleration may synergistically contribute to their enhanced intrusion into LMEs situated along the paths of these boundary currents, further leading to the intensification of persistent fronts.” This is an important claim, but it's not supported by any evidence. Why would boundary current warming lead to changes in the fronts within the boundary currents themselves?

Lines 234-258: Similarly to the above, I can't really understand why this paragraph is in this paper. The authors have shown evidence of changes in persistent fronts in boundary current regions, how does this finding link to changes in coastal upwelling systems, and why would these changes “be attributed to both the enhanced upwelling and the buffer effects of upwelling regions”? No evidence for this statement is provided.

Lines 262-264: “It is important to 263 acknowledge the relatively high uncertainty in satellite-observed SST data in polar regions with high sea ice concentrations” I would also note that SST data in this region is seasonally biased, being far more reliable in summer than winter.

Line 266: “This reanalysis dataset was derived from an eddy-resolving (~9 km)” In high latitudes and in stratified environments, such as the Arctic, the Rossby radius of deformation is small (5-10km) and 9km grid spacing is unlikely to resolve all but the largest mesoscale eddies, and none of the sub-mesoscale. See Hallberg 2013 and Nurser and Bacon (2014).

Fig. 1: I’m afraid I can’t read the axes of panels b) and c). I’d make each of these panels as wide as the page and stick them vertically.

Nurser, George & Bacon, S.. (2014). The Rossby radius in the Arctic Ocean. *Ocean Sci.* 10. 967-975. [10.5194/os-10-967-2014](https://doi.org/10.5194/os-10-967-2014).

Hallberg (2013), Using a resolution function to regulate parameterizations of oceanic mesoscale eddy effects, *Ocean Modelling* (72)

Graham, R. M., A. M. deBoer, K. J. Heywood, M. R. Chapman, and D. P. Stevens (2012), Southern Ocean fronts: Controlled by wind or topography?, *J. Geophys. Res.*, 117, C08018, doi:10.1029/2012JC007887.
Chapman, C.C., Lea, MA., Meyer, A. et al. Defining Southern Ocean fronts and their influence on biological and physical processes in a changing climate. *Nat. Clim. Chang.* 10, 209–219 (2020).
<https://doi.org/10.1038/s41558-020-0705-4>

Wunsch, C., Williamson, S., and Heimbach, P.: (2023) Potential artifacts in conservation laws and invariants inferred from sequential state estimation, *Ocean Sci.*, 19, 1253–1275, <https://doi.org/10.5194/os-19-1253-2023>, 2023.

Reply to Reviewer # 1

Reviewer #1 (Remarks to the Author):

Review of the manuscript NCOMMS-23-62487-T entitled “Global mapping and evolution of persistent fronts in large marine ecosystems over the past 40 years” by Qinwang Xing, Haiqing Yu, and Hui Wang submitted to Nature Communications

I reviewed two previous papers of the same lead author and made several major comments on methods that have been addressed. Therefore, in this review I’m not going to dwell on methods even though the authors made a few modifications. After all, the authors made their front detection code available, which is commendable. Overall, this work is a major contribution, and it should be published pending minor fixes listed below.

A: We would like to thank you for your time and effort in reviewing our manuscript and for providing positive comments. We have incorporated all of your suggestions into our revisions, please see the detailed reply as follows.

Minor comments:

Q1: Line 1-2 and similar instances throughout the paper: “Large Marine Ecosystems” is a standard term, with each word capitalized.

A: Thank you for your thorough review. We have corrected them in the revised manuscript (Line 1-2).

Q2: L61: “topographical change” should be “bathymetric gradients”

A: Thank you for your suggestion. We have replaced “topographical change” by “bathymetric gradients” (Line 60).

Q3: L62: “melting sea ice” – Consider “sea ice freeze-thaw”

A: Thank you for your suggestion. We have replaced “melting sea ice” by “sea ice freeze-thaw” (Line 61).

Q4: L126: “cover of glacier” should be “sea ice cover”

A: Thank you for your suggestion. We have replaced “cover of glacier” by “sea ice cover” (Line 134).

Q5: L131: “Occurrence and intensity, indicating” should be “Occurrence and intensity defined, respectively, as”

A: Thank you for your suggestion. We have implemented this modification (Line 139).

Q6: L163: “boundary current region” should be “regions”

A: Thank you for your suggestion. We have replaced all “region” by “regions” (Line 176).

Q7: L209: “around encompassing” should be just “encompassing”

A: Thank you for your suggestion. We have removed “around” (Line 220).

Q8: L216: “West boundary currents” should be “Western boundary currents”

A: Thank you for your suggestion. We have replaced “West” by “Western” (Line 229).

Q9: L216: “such as the Kuroshio current, Gulf Stream, and Agulhas current” – Consider listing all WBCs.

A: Thank you for your suggestion. We have incorporated all western boundary currents in this sentence: “*such as the Kuroshio current, Gulf Stream, Agulhas current, East Australian current, and Brazil/Malvinas current*” (Line 229-230).

Q10: L342: “contract their suitable habitats and distribution ranges poleward” should be “expand their distribution ranges poleward”

A: Thank you for your suggestion. We have implemented this modification, as follows: “*Ocean warming and consequent deoxygenation compel marine species to expand their distribution ranges poleward*” (Line 356-357).

Q11: L343: Delete “climate-driven reductions in fish catch” altogether because this statement is false. Many species, especially those that inhabit cold waters, thrive in a warmer climate. For example, the population and catches of some salmonids in the North Pacific exploded lately. Cod and capelin returned to the Barents Sea. Haddock and cod in the NW Atlantic have mostly recovered after the collapse of the 1980s caused in part by extremely cold winters in the early 1980s. Farther south, winter flounder’s stock off New England collapsed after two consecutive extremely harsh winters in the mid-1960s; the stock recovered thanks to the subsequent warming. Herring and cod in the Norwegian Sea are doing very well in a warmer climate. All of the above is well documented in the literature. These are just a few examples. Their number can be multiplied. Moreover, fish catches depend on a variety of factors; climate is just one of many factors. Such factors as overfishing or regulations (quotas, restrictions etc.) have a much greater impact on fish catches than climate variability except for such phenomena as El Niño.

A: Thank you for your comment, and we completely agree with your perspective. We have replaced “climate-driven reductions in fish catch” by “climate-driven change in maximum catch potential” (Line 358). This statement is derived from Vicky et al., 2020 (*Nature Reviews Earth & Environment*, <https://doi.org/10.1038/s43017-020-0071-9>). In addition to the periodic fluctuations in the capture of certain fishes that you mentioned, some studies have also demonstrated the long-term increase or decrease in fish stocks related to global warming (Vicky et al., 2020). The revised sentence now reads, “*Aside from the impacts of overfishing or inadequate regulations, this climate-driven change in maximum catch potential can have adverse socio-economic consequences, particularly for tropical countries*” (Line 357-360).

Q12: L345: “the unclear trends for persistent fronts in tropical regions may exacerbate the vulnerability of countries” – It is unclear to me how “unclear trends” may “exacerbate” anything.

A: Thank you for your comment. This sentence contained an incorrect statement,

and it has been modified to: *“the decreasing trends for persistent fronts in some tropical regions may exacerbate the vulnerability of countries”* (Line 361-362).

Q13: L356: “This has become increasingly urgency...” – urgent not urgency.

A: Thank you for your suggestion. We have replaced “urgency” by “urgent” (Line 372).

Q14: L369: “carbon export” should be “carbon sequestration by the ocean”

A: Thank you for your suggestion. We have implemented this modification (Line 384).

Q15: L562: “boundary current region” must be regions.

A: Thank you for your suggestion. We have replaced “region” by “regions” (Line 588).

Q16: L702: “n/a-n/a” – Complete the reference.

A: Thank you for your thorough review. We have double-checked and completed all references accordingly.

Reply to Reviewer # 2

Reviewer #2 (Remarks to the Author):

This study presents a global, fine-scale digital atlas of persistent ocean fronts derived from satellite observations and demonstrates statistically significant increase in the occurrence or intensification of persistent ocean fronts, particularly in the subtropics and Arctic regions.

The idea of a fine-scale digital atlas of persistent fronts itself is not actually very unique, as the same approach has been done in Mauzole (2022), who produced a global distribution of persistent fronts using satellite SST fields in association with large marine ecosystem (LME). The central novel aspect of this study, as the authors point out, is the global analysis of long-term trends in the ocean front, which are not very

well captured by reanalysis and high-resolution ocean models. Given the crucial role of ocean fronts in ecosystem as well as in the climate system, this study suggests significant implications for various applications.

However, I find that the figures do not efficiently convey the results, and the introduction could benefit from a more refined structure to enhance clarity and engagement. I encourage the authors to make major revisions to the text, especially the introduction part, and figures with a focus on improving clarity and context to better highlight the long-term trend. Some more specific comments are listed below.

A: Thank you for taking the time and making the effort to review our manuscript. We have revised the Introduction section and some figures based on your suggestion, and all of your suggestions have been incorporated into the revision. Please find a detailed reply below.

Major comments:

1. Introduction

Q1: 1.1 The context of the introduction is rather discursive, but could be improved by clarifying the points of each paragraph. For example, the first paragraph (especially L49-56) and the second paragraph seem to repeat the physical nature and influence of ocean fronts.

A: Thank you for your insightful comment. We have removed the mentioned sentences from the first paragraph of the manuscript to avoid repetition. Additionally, we have rewritten or added transitional sentences at the beginning or end of each paragraph to enhance the coherence and logic of the Introduction. Currently, the revised Introduction follows this logical order:

Paragraph 1: Introducing the subject of long-term variations in discrete oceanographic features under the background of large-scale oceanic change.

Paragraph 2: Stating the climatic and ecological significance of fronts and introducing persistent fronts.

Paragraph 3: Discussing the challenges associated with studying the long-term changes in persistent fronts.

Paragraph 4: Introducing the focus of this manuscript in addressing this challenge.

Q2: 1.2 It is unclear what is meant by a persistent front, although the authors briefly mention it as “sufficiently long-lived and persistent front” in L67-68. Since the thesis of this study emphasizes the trend of persistent fronts as the main finding, an elaboration of the definition of persistent front in plain language (not about the detailed detection algorithm) would help readers to better understand what this study is for.

Regarding the trend, in addition, I also wonder if the trend of frontal occurrence is sensitive to the choice of threshold of “persistence”. Or does this study adopt a widely-used threshold definition or something?

A: Thank you for your good suggestion. We have incorporated the definition of persistent fronts in simpler language (Line 70-72), as follows: “persistent fronts, also called quasi-stationary fronts (defined as a unique frontal aggregation where fronts frequently occur over relatively long periods)”. The threshold for “persistence” is set at 1% to generate persistent front fields, aligning with the value employed in previous studies (Mauzole 2022; Xing et al., 2023). The choice of this threshold should be understood as a compromise for the method to work globally to balance between frontal features and background noise (Mauzole 2022). Meanwhile, in the revision, we have incorporated some sensitivity analyses. Comparisons between the results obtained with a 1.0% threshold and those with thresholds of 0.7% and 1.3% suggest that the long-term changes in persistent fronts are not sensitive to the selection of the threshold value (see Supplementary Fig. 11).

Supplementary Fig. 11 | Sensitivity analysis of threshold values for persistent frontal changes. a, b, Linear trend comparisons of the occurrence and intensity of persistent

fronts between the results obtained with a 1.0% threshold and those with thresholds of 0.7% (a) and 1.3% (b) in each LMEs.

Mauzole, Y. L. (2022). Objective delineation of persistent SST fronts based on global satellite observations. *Remote Sensing of Environment*, 269, 112798.

Xing, Q., Yu, H., Wang, H., & Ito, S. I. (2023). An improved algorithm for detecting mesoscale ocean fronts from satellite observations: Detailed mapping of persistent fronts around the China Seas and their long-term trends. *Remote Sensing of Environment*, 294, 113627.

Q3: 1.3 The authors argue that the global atlas of persistent fronts in LMEs has not been reported based on objective algorithm (L93-95). However, I find that the idea and approach of this study actually follows Mauzole (2022), although the detailed algorithm may differ. Rather, the novelty of this study comes from the verification of the long-term trend of persistent fronts derived from 40 years of global satellite observations including the Arctic region. In this sense, overviews of global changes in LMEs and fronts should be further addressed to strengthen the context regarding the contribution of this study.

A: Thank you for your professional comment. We acknowledge that our previous unclear expression may have caused some misunderstandings. The automated algorithm developed by Mauzole (2022) is designed to obtain global persistent front fields (such as Supplementary Fig. 1), rather than creating a global atlas of persistent fronts with a specific emphasis on depicting each individual persistent front, as demonstrated in Belkin et al. (2009) where each front is separately extracted and named. The persistent front fields in Mauzole (2022) represent potential persistent frontal pixels globally, which may be discretely or continuously located in various local areas. However, these fields cannot be utilized to assess the long-term changes of each individual persistent front that are composed of a series of adjacent pixels. In our study, we have established a detection framework for persistent fronts based on four seasonal persistent front fields, putting significant effort into creating an unprecedented

global digital atlas of each individual persistent front. This atlas provides detailed information on the names, locations, and seasonal patterns of 1,198 identified persistent fronts. Readers can select any persistent fronts they are interested in from this atlas to further investigate the dynamic, ecological effects, and long-term evolution of these persistent fronts. The persistent front fields in Mauzole (2022) cannot provide this information and potential application. Another significant difference is that our method incorporates a recently improved front detection algorithm (Xing et al., 2023), capable of identifying many coastal persistent fronts that might be missed by the method used in Mauzole (2022). This work represents the inaugural application of the improved algorithm to the global oceans. Therefore, the novelty of this study lies in both the creation of an unprecedented global digital atlas of persistent fronts and the exploration of their long-term trend global pattern, and we have revised this part of the Introduction to clarify this point, as follows: *“However, although existing research has presented global persistent front fields based on the relatively objective and automatic delineation algorithm²⁹, a comprehensive global atlas with a specific focus on depicting each individual persistent front in LMEs, such as each separately extracted and named front in ref.¹², has not been systematically censused and updated using these persistent front fields. This absence makes it difficult for us to accurately evaluate the long-term changes of each individual persistent front in a changing climate”* (Line 97-103).

Meanwhile, based on your suggestion, we have also incorporated some overviews about frontal change to strengthen the context regarding the contribution of this study, as follows: *“Although previous studies have provided examples of frontal variations in some regional oceans, such as the meridional shifts of Southern Ocean fronts^{22,23}, the southwestward displacement of Patagonian shelf break front²⁴, and the meridional shift of the Oyashio extension front²⁵, our understanding of how global fronts will change in term of occurrence and intensity under climate change remains limited, particularly for the aforementioned persistent fronts”* (Line 78-83).

Belkin, I. M., Cornillon, P. C., & Sherman, K. (2009). Fronts in large marine ecosystems.

Progress in Oceanography, 81(1-4), 223-236.

Mauzole, Y. L. (2022). Objective delineation of persistent SST fronts based on global satellite observations. *Remote Sensing of Environment*, 269, 112798.

Xing, Q., Yu, H., Wang, H., & Ito, S. I. (2023). An improved algorithm for detecting mesoscale ocean fronts from satellite observations: Detailed mapping of persistent fronts around the China Seas and their long-term trends. *Remote Sensing of Environment*, 294, 113627.

2. Clarity of figures

Q4: 2.1 LME is an important keyword in this study. However, its specific horizontal distribution is shown in Fig. 2, while the regional analysis of fronts in each LMEs is presented in Fig. 1b,c. This figure composition leads to confusion such that the authors explain “LME64” (L125) and “LME 20” (L127) before showing what they are (although the authors refer to Extended Data Fig. 3, it is an inefficient repetition since the same information is presented in Fig. 2). Reorganizing the figures may help to better structure the manuscript.

A: Thank you for your suggestion. We have reorganized Figure 1 based on the input from Reviewer #3 to enhance readability. Additionally, we agree that displaying the horizontal distribution of LMEs before the analysis results of each LME would be beneficial. Therefore, in Figure 1a, we have included LME number characters in their respective distributions. These characters are sufficiently large for readers to easily identify them. While Supplementary Fig. 3 (originally named Extended Data Fig. 3) also presents LME numbers, we believe its inclusion is essential rather than repetitive. This figure outlines the classification of various latitudinal regions and the names of each LME, forming the basis of our analysis. Our study categorizes LMEs into different groups based on distinct latitudinal regions to facilitate statistical analysis of persistent front changes across these categories. This approach aids in identifying global patterns of persistent frontal changes. The inclusion of LME number characters in Figures 1 and 2 aims to assist readers in referring to the names of LMEs they are interested in by consulting Supplementary Fig. 3.

Q5: 2.2 Tiny characters and numbers in Figs. 1,2 are hard to discern. I would recommend the authors to consider a more efficient way to leave key information and trim less important things, such as ID numbers of fronts, by moving them to Supplementary figures.

A: Thank you for your suggestion. In the revised Figure 2, we have significantly increased the size of the characters representing LME numbers to enhance clarity. Additionally, in Figure 1, the tiny characters and numbers represent the ID numbers of each persistent front and the LME numbers. We have substantially increased the size of the LME number characters in the revised Figure 1 to improve visibility. However, after careful consideration, we decide not to remove but slightly increase the size of the ID number characters of each persistent front. We believe these ID numbers are crucial for this study for two reasons: firstly, they contribute to highlighting the distinction (Please refer to the detailed explanation in the response to Major comments Q3 regarding the distinction between persistent frontal fields and the maps of persistent fronts) between our persistent front atlas and the persistent front fields of Mauzole (2022); secondly, some readers may be interested in specific persistent fronts and can find more information about them in Table S1 based on their ID numbers. The large size of ID number characters may lead to mutual occlusion and reduced visibility, especially given the numerous identified persistent fronts. Consequently, in the revision, we have manually checked and moved each ID number character one by one to ensure that each character is clear and visible, and we have also provided high-resolution figures to ensure the visibility of ID numbers. Interested readers can enlarge Figure 1 to view the ID numbers of their selected persistent fronts.

Q6: 3. Figure 4 presents the results from two models. Although the result of the reanalysis is shown as a horizontal map in the Extended Data figure, expanding Figure 4 to include the reanalysis would clarify the usefulness of the global atlas openly provided in this study.

A: Thank you for your suggestion. We have incorporated the result of this

reanalysis data in Figure 4 in revision (Line 407).

Minor comments:

Q7: Reference 3 seems to be incorrect.

A: Thank you for your thorough review. We have corrected it in the revision.

Q8: L118-120: It may be unfair to compare the number of persistent fronts detected in this study with outdated hand-drawn maps, despite the significant improvements in recent studies.

A: Thank you for your comment. We apologize for the incorrect citation in the original manuscript. We have compared our results with Belkin et al. (2009). While the global hand-drawn maps of persistent fronts were presented in Belkin et al. (2009), we believe these maps are not outdated. This influential paper has been widely cited in many studies, and its hand-drawn locations of persistent fronts have been extensively used in previous research. Although global persistent front fields have been displayed using automated detection algorithms in recent studies, such as Mauzole (2022), up to now, the hand-drawn maps of Belkin et al. (2009) remain irreplaceable as they provide detailed information on the names and locations of each individual persistent front. Please refer to the detailed explanation in the response to Major comments Q3 regarding the distinction between persistent frontal fields and the maps of persistent fronts. Therefore, our global atlas of persistent fronts can only be compared with these hand-drawn maps, and no other map is more suitable. We have included the phrase “*that have been extensively cited and used in previous research*” in this sentence to provide clarification (Line 126-127).

Q9: L121: Please elucidate what the magnitude of the increase in frontal occurrence as much as 11 indicates (elaborate it in terms of physical meaning, not just adding its unit).

A: Thank you for your comment. We have revised this sentence to clarify it, as follows: “*The number of identified persistent fronts has significantly increased in most*

LMEs compared to that of existing hand-drawn maps, with an average increase of 11 persistent fronts” (Line 127-128).

Q10: Fig 1a caption: meaning of blue shading is missing

A: Thank you for your thorough review. We have included the statement “*Blue shadings represent the domain of each LME*” in the revised version (Line 168-169).

Q11: L167: Please specify a reference of Mann-Kendall test and Sen’s slope estimator. This description of statistics may be moved to Method section.

A: Thank you for your comment. This sentence has been moved to Method section, and its reference has been added in the revision (Line 572-575).

Q12: In Fig. 2b,d, please specify the left and right bars in the caption. Also in L202, specify black bars are indicating error bars, because there are another black bars in the figure.

A: Thank you for your thorough review. We have included the sentence “*while the bar charts with dark borders and without borders represent their linear trend and relative trend, respectively*” to provide clarity (Line 212-213). Additionally, the issue regarding error bars has been corrected in the revised version.

Q13: L254: Here, do the authors indeed mean “decreasing trend” of upwelling? Or cooling trend? It’s confusing because the next sentence indicates enhanced upwelling.

A: Thank you for your thorough review. We have corrected the term to “cooling trend” in the revised version (Line 267). This cooling trend signifies enhanced upwelling.

Q14: L358: intensity => intensify?

A: Thank you for your comment. This word has been revised to “enhance” (Line 373).

Q15: I don’t see black lines in Extended Data Fig. 1.

A: Thank you for your thorough review. It has been replaced with a high-resolution figure in the revision.

Reply to Reviewer # 3

Reviewer #3 (Remarks to the Author):

This paper develops a global dataset of persistent SST fronts in Large Marine Ecosystems (LMEs), typically confined to over and near continental shelves. The authors identify persistent changes in fronts, including changes in location, frequency of occurrence and intensity. Changes are particularly notable in the Arctic ocean, which the authors link to changes in sea ice concentration.

For the most part, this is a fine study with a relatively straightforward analysis. However, I have some concerns about various statements made within the paper for which no evidence it provided. Particularly, the authors attempt to attribute changes in fronts in boundary current regions to changes in upwelling and boundary currents themselves. While both these points are indeed possible, they need to be supported with evidence. I've also noted a few other issues that should be relatively easy to fix or discuss. However, several revisions to the manuscript are required before I could recommend publication.

A: Thank you for taking the time and effort to review our manuscript. We have carefully addressed your concerns about statements without support by either deleting unimportant information or adding relevant discussions. Additionally, we have implemented all your suggestions in the revision. Please find a detailed reply below.

Major points

Q1: Statements without supporting evidence

As mentioned above, there are several statements made by the authors without evidence. I've pointed out two in detail below in particular comments, but there may be others. I would encourage the authors to evaluate their manuscript critically, and ensure that all positive statements are supported.

A: Thank you for your comment. We have addressed the examples that you

pointed out by either deleting unimportant statements or adding relevant discussions; please see the responses in Q10 and Q11. Meanwhile, we have also gone over and revised the entire manuscript to ensure that all positive statements are supported.

Q2: LME Definition

How have the authors defined a Large Marine Ecosystem? There are various, occasionally conflicting, definitions in the literature, and while the authors have cited several references, it is unclear to me where the regions covered by LME have come from. This should be explicit.

A: Thank you for your comment. In collaboration with the University of Rhode Island, NOAA introduced the LMEs concept over 30 years ago. This model serves as a framework for implementing ecosystem approaches in the assessment, management, recovery, and sustainability of resources and environments within LMEs. As ecosystem-based management units, the designation of LMEs represents an ongoing scientific process (Sherman, 1991), and the number and boundaries of LMEs may evolve over time. For example, the total number of LMEs has changed from 64 to 66 due to the splitting of previous Arctic LMEs (PAME, 2013). For this study, we utilized the currently identified 66 LMEs developed by the NOAA Large Marine Ecosystems Program (Sherman et al., 2016). This information has been incorporated into the Method section in the revision, as follows: *"It should be noted that, as ecosystem-based management units, the designation of LMEs represents an ongoing scientific process⁷⁶, and the number and boundaries of LMEs may evolve over time. For this study, we utilized the currently identified 66 LMEs developed by the NOAA Large Marine Ecosystems Program⁷⁷"* (Line 589-593).

PAME. (2013). Large Marine Ecosystems (LMEs) of the Arctic area: revision of the Arctic LME map 15th of May 2013. Akureyri, Iceland: Conservation of Arctic Flora and Fauna (CAFF) and Protection of the Arctic Marine Environment (PAME).

Sherman, K. (1991). The large marine ecosystem concept: research and management strategy for living marine resources. *Ecological Applications*, 1(4), 349-360.

Sherman, K., & Hamukuaya, H. (2016). Sustainable development of the world's Large Marine Ecosystems. *Environmental Development*, 17, 1-6.

Q3: Use of Reanalysis Output for analysis of trends

In this paper, the authors use the output of an ocean reanalysis in ice covered regions. While the reanalysis generally supports the satellite observations, it should be noted that reanalysis products are subject to errors in trends due to changes in the observing systems assimilated into them. For example, the authors use reanalysis output from 1993-onwards, but the availability of Argo floats from ~2005 results in trends in the reanalysed output that are not due to “true” changes in the system, but due to Argo coming online and constraining the model. See Wunsch

While the authors can't exactly re-run the reanalysis, discussion of these issues should at least be noted and discussed. Trends from reanalysis can be highly unreliable, and sea-ice would be more difficult to interpret still, given the lack of observations of sea-ice thickness.

Wunsch, C., Williamson, S., and Heimbach, P.: (2023) Potential artifacts in conservation laws and invariants inferred from sequential state estimation, *Ocean Sci.*, 19, 1253–1275, <https://doi.org/10.5194/os-19-1253-2023>, 2023.

A: Thank you for your comment. We completely agree with your point of view that changes in the observing systems may result in the bias of reanalysis. This point did not affect our results as our findings are mainly based on satellite observations, and our findings of global variations in persistent fronts provide a baseline for refining high-resolution ocean models. Therefore, based on your suggestion, we have added some discussion as follows: *“It should be noted that the potentially time-varying bias in this reanalysis dataset may lead to an unreliable trend for SST and persistent fronts⁶⁸, despite being based on one of the most advanced ocean models incorporating observational assimilation and having been widely applied in previous studies. For example, the rapid increased availability of Argo floats since the 21st century may result in trends that are not due to true changes in the system, and the absence of sea ice thickness in the observational assimilation may make it challenging to accurately*

depict changes in polar oceans” (Line 445-452).

Particular comments

Q4: Line 32: “rapid increases” and “slight decreases” of what? Occurrence, intensity, both?

A: Thank you for your comment. It refers to both occurrence and intensity of persistent front. We have included this information in the revision, as follows: “*In subtropical regions around boundary currents and upwelling systems, as well as in polar regions, the occurrence and intensity of persistent fronts exhibit a rapid increase, while remaining stable or experiencing a slight decrease in tropical regions*” (Line 31-34).

Q5: Line 67: I'd probably state “phytoplankton blooms” or just “plankton blooms”, as opposed to “primary production blooms”.

A: Thank you for your comment. We have replaced “primary production blooms” by “phytoplankton blooms” (Line 66).

Q6: Line 76-77: “our understanding of how they will change under global warming remains limited.” I don't agree with this statement. There is a huge amount of work, particularly in the Southern Ocean, on how fronts will change in the future. See Chapman et al. 2020 for a review, and Graham et al

Graham, R. M., A. M. deBoer, K. J. Heywood, M. R. Chapman, and D. P. Stevens (2012), Southern Ocean fronts: Controlled by wind or topography?, *J. Geophys. Res.*, 117, C08018, doi:10.1029/2012JC007887.

Chapman, C.C., Lea, M.A., Meyer, A. et al. Defining Southern Ocean fronts and their influence on biological and physical processes in a changing climate. *Nat. Clim. Chang.* 10, 209–219 (2020). <https://doi.org/10.1038/s41558-020-0705-4>

A: Thank you for your comment. Based on the valuable references you provided, we have revised this sentence as follows: “Although previous studies have provided examples of frontal variations in some regional oceans, such as the meridional shifts

of Southern Ocean fronts^{22,23}, the southwestward displacement of Patagonian shelf break front²⁴, and the meridional shift of the Oyashio extension front²⁵, our understanding of how global fronts will change in term of occurrence and intensity under climate change remains limited, particularly for the aforementioned persistent fronts” (Line 78-83).

Q7: Line 126: “where fronts cannot be found due to the cover of glacier all year round” do the authors mean sea-ice instead of glaciers?

A: Yes, you are right. Thank you for your thorough review. We have replaced “glacier” by “sea ice” (Line 134).

Q8: Line 139: “The frequent presence of sea ice may weaken the formation and identifiability of persistent fronts” Could it also be said that the presence of sea-ice prevents measurement of SST from space? Sea-ice zones, even those are only seasonally impacted, lose large amount of data simply due to high levels of sea-ice in the winter, and contamination during periods when only low concentrations of sea-ice are present.

A: Yes, you are right. we have added this information to make it clear, as follows: *“The presence of sea ice prevents the measurement of SST from space, making it difficult to detect potential fronts beneath the sea ice”* (Line 148-149).

Q9: Line 145-147: “We selected typical LMEs around boundary currents and upwelling systems and found that persistent fronts exhibit high occurrence and intensity, with an average occurrence of 4.43% and 4.16%” The authors suggest here that an occurrence of 4.4% is “high”. However, that’s compared with a global average of 3.4%. If we translate this to number of days with a front per year, that’s 16.2 days/year vs a global average 12.6 days/year. While higher than the global average, this doesn’t strike me as particularly “high” and probably only borderline statistically significant.

A: Thank you for your comment. We believe that the value “16.2 days/year” and “12.6 days/year” from your calculation are not true because frontal occurrence is

determined by counting the number of frontal pixels within a 6-pixel radius (~30 km) of persistent fronts and dividing this by the total number of available SST pixels within the same distance, rather than only counting the frontal pixels located in the persistent fronts. This area-average definition considers their potential location moving in different seasons. The LME-integrated result of frontal occurrence, together with the area-average definition, may narrow the difference between high and low frontal occurrence, making a significant difference seem less pronounced. Therefore, we have added some sentences to clarify it, as follows: *“Although their average occurrence is only ~1% higher than the global average, considering the area-average definition (see Method) and LME-integrated calculation for persistent frontal occurrence, this difference is sizable and not negligible”* (Line 159-161).

Q10: Line 229-233: “The combined effects of path alterations, broadening of boundary currents, and their rapid warming and acceleration may synergistically contribute to their enhanced intrusion into LMEs situated along the paths of these boundary currents, further leading to the intensification of persistent fronts.” This is an important claim, but it’s not supported by any evidence. Why would boundary current warming lead to changes in the fronts within the boundary currents themselves?

A: Thank you for your comment. We acknowledge that there is a logical error in this point. Our original intention was to convey that the temperature difference of fronts separating the boundary current and surrounding water masses would be enhanced by warming boundary currents. However, due to the lack of direct evidence and its minimal impact on our findings, we have deleted the point in the revised version (Line 243-246).

Q11: Lines 234-258: Similarly to the above, I can’t really understand why this paragraph is in this paper. The authors have shown evidence of changes in persistent fronts in boundary current regions, how does this finding link to changes in coastal upwelling systems, and why would these changes “be attributed to both the enhanced upwelling and the buffer effects of upwelling regions”? No evidence for this statement

is provided.

A: Thank you for your comment. It's possible that our unclear expression may have caused some confusion. We want to emphasize that this paragraph has a crucial distinction from the previous one. In our study, we identified a notable intensification of persistent fronts around both boundary currents and upwelling systems within subtropical regions (Fig. 2). Considering the different dynamic processes between boundary currents and upwelling systems, we further discussed potential dynamic mechanisms in the two types of regions, respectively. This paragraph discussed potential reasons of strengthening persistent fronts in the famous upwelling systems, while previous paragraph that you mentioned discussed reasons in boundary currents. Therefore, to enhance reader comprehension and facilitate a smooth transition between topics, we have added a sentence to clarify the flow of the discussion, as follows: "*Considering the different dynamic processes between boundary currents and upwelling systems, we further discussed potential dynamic mechanisms in the two types of regions, respectively*" (Line 226-228).

Based on the aforementioned structure of this paragraph, we discuss how global warming strengthens upwelling and fronts by increasing upwelling-favorable winds, supported by extensive evidence from previous research. The contrast between cold water masses from upwelling and the surrounding warm water masses generates fronts with a strong cross-shore SST gradient in upwelling systems. The increase in upwelling-favorable winds brings more cold deeper waters to the surface oceans, combined with climate change-related warming of the surrounding ocean, contributing to the intensification of persistent fronts in upwelling systems. Meanwhile, our additional analysis of the long-term trend of SST around upwelling systems also supports this discussion (Supplementary Fig. 7). Therefore, we believe that the enhanced upwelling may play a key role in intensifying persistent fronts in upwelling systems (the buffer effects have been removed in the revision) (Line 247-271).

Q12: Lines 262-264: "It is important to acknowledge the relatively high uncertainty in satellite-observed SST data in polar regions with high sea ice concentrations" I would

also note that SST data in this region is seasonally biased, being far more reliable in summer than winter.

A: Thank you for your reminder. We have incorporated this information into this sentence as follows: *“It is important to acknowledge the relatively high uncertainty in satellite-observed SST data in polar regions with high sea ice concentrations, where SST is generally more reliable in summer than in winter”* (Line 275-276). This is the reason why we conducted the analysis of Fig. 3 using only the data from summer.

Q13: Line 266: “This reanalysis dataset was derived from an eddy-resolving (~9 km)”
In high latitudes and in stratified environments, such as the Arctic, the Rossby radius of deformation is small (5-10km) and 9km grid spacing is unlikely to resolve all but the largest mesoscale eddies, and none of the sub-mesoscale. See Hallberg 2013 and Nurser and Bacon (2014).

Nurser, George & Bacon, S.. (2014). The Rossby radius in the Arctic Ocean. *Ocean Sci.* 10. 967-975. 10.5194/os-10-967-2014.

Hallberg (2013), Using a resolution function to regulate parameterizations of oceanic mesoscale eddy effects, *Ocean Modelling* (72)

A: Thank you for your comment. We provided an incorrect resolution for this reanalysis dataset in the original manuscript. The reanalysis dataset has a resolution of $1/12^\circ$ (9.25 km at the equator and around 4.5 km at subpolar latitudes), which essentially meets the requirements of the Rossby radius of deformation. We have corrected this information in the revision (Line 280).

Q14: Fig. 1: I’m afraid I can’t read the axes of panels b) and c). I’d make each of these panels as wide as the page and stick them vertically.

A: Thank you for your good suggestion. We have reorganized the Figure 1 based on your suggestion (Line 163).

REVIEWER COMMENTS

Reviewer #2 (Remarks to the Author):

The authors have adequately addressed my concerns in the revised manuscript, and I would like to recommend acceptance of this study for Nature communications after correcting a few minor issues.

- 1) L105: "LMEs who ..." does not seem appropriate
- 2) The meaning of circled numbers in Fig. 1 is missing from the figure caption.

Reviewer #3 (Remarks to the Author):

The authors have done a good job in responding to my concerns and modifying the manuscript where appropriate, and I thank them for their efforts. I'm now happy to recommend publication subject to one more major-ish comment below, where I think the authors do their work (and themselves) a disservice by not using their results to their full potential.

Section Intensified persistent fronts along boundary currents and upwelling

I find this section to be very speculative, and much of it reads like an introductory section, describing the current state-of-the-art. The links to the results found by the authors are tenuous, and left to just a couple of short sentences.

Could the authors put some more emphasis into describing their results in these important regions? Are the persistent fronts moving poleward? Is it clear in the major WBC or upwelling systems that there has been an increase in the SST gradient between the inshore and offshore systems? Are increasing gradients limiting cross shelf exchange processes? The authors have the tools to delve into these questions in detail, rather than a very general discussion of perviously published studies.

Lines 44-45: I'd also update reference #4 to something more recent (see Li et al. 2023), which states that Western Boundary Currents are unlikely to be intensifying, but changing state.

Line 65: I think it's worth noting that fronts can transport heat out of the ocean, as well as into the interior.

Lines 135-138: Are the large numbers of persistent fronts identified in the named LMEs simply due to the fact that those LMEs cover larger areas (such as the Antarctic LME which is circum-hemispheric)? Could some measure of fronts per unit area be derived?

Line 225: As above, the paper(s) by Junde Li and collaborators could be useful to cite here, as they've specifically studied the dynamic response of boundary currents to climate change.

Lines 232-233: WBCs also tend to be highly eddying, and the interaction of eddies with the coast, the boundary currents, and other eddies, tends to lead to the formation of fronts;

Li, J., Roughan, M., & Kerry, C. (2021). Dynamics of interannual eddy kinetic energy modulations in a Western Boundary Current. *Geophysical Research Letters*, 48, e2021GL094115.

<https://doi.org/10.1029/2021GL09411544->

Reviewer #3 (Remarks on code availability):

Code requires more comments and a brief document explaining what it does and how it can be run. Variable names (for example, 'mm') are not clear, which makes the code difficult to follow.

It appears that the code requires use of the image-processing toolbox, which is not standard in Matlab, and could potentially stop an interested user from applying the code.

Reply to Reviewer # 2

Reviewer #2 (Remarks to the Author):

The authors have adequately addressed my concerns in the revised manuscript, and I would like to recommend acceptance of this study for Nature communications after correcting a few minor issues.

A: Once again, we would like to thank you for recognizing our work and for your constructive comments. We have addressed all of the comments in the revision.

Q1: L105: "LMEs who ..." does not seem appropriate

A: Thank you for your comment. We have removed these words in the revision (Line 105).

Q2: The meaning of circled numbers in Fig. 1 is missing from the figure caption.

A: Thank you for your comment. We have incorporated its meaning into our manuscript as follows (Line 172-173): "*while the overlapping circled numbers within each LME denote the LME numbers*".

Reply to Reviewer # 3

Reviewer #3 (Remarks to the Author):

The authors have done a good job in responding to my concerns and modifying the manuscript where appropriate, and I thank them for their efforts. I'm now happy to recommend publication subject to one more major-ish comment below, where I think the authors do their work (and themselves) a disservice by not using their results to their full potential.

A: Thank you for taking the time and effort to review our manuscript once again. We have addressed these issues in the revision.

Q1: Section Intensified persistent fronts along boundary currents and upwelling

I find this section to be very speculative, and much of it reads like an introductory section, describing the current state-of-the-art. The links to the results found by the authors are tenuous, and left to just a couple of short sentences.

Could the authors put some more emphasis into describing their results in these important regions? Are the persistent fronts moving poleward? Is it clear in the major WBC or upwelling systems that there has been an increase in the SST gradient between the inshore and offshore systems? Are increasing gradients limiting cross shelf exchange processes? The authors have the tools to delve into these questions in detail, rather than a very general discussion of perviously published studies.

A: Thank you for your comment. The aim of this section is to determine the enhancement of persistent fronts in boundary currents and upwelling systems and to discuss potential mechanism. In earlier sections, we mapped out persistent fronts and identified hotspots of intensified fronts in subtropical and polar regions. Subsequently, we focused on representative LMEs around boundary currents and upwelling systems to illustrate their long-term changes in persistent fronts. We then conducted a comprehensive discussion on the formation of persistent fronts and potential reasons for their intensification, drawing from previous studies. Therefore, we believe that this section is logically coherent and is closely related to our result. Actually, the novelty of our study lies in both the creation of an unprecedented global digital atlas of persistent fronts and the evidence of their long-term trend global pattern, which are not well captured by high-resolution reanalysis and Earth system models. The majority of the content in this section and subsequent sections should be considered as reasonable discussion further based on our findings. It reads a bit like an introductory section, but this is due to our intention to provide more detailed statements to ensure readers unfamiliar with this field can easily follow along. This makes our manuscript better meet the requirements of the wide audience of this comprehensive journal.

We also greatly appreciate your suggestions. Your insights regarding the poleward shift of persistent fronts and the effects of intensified fronts on cross-shelf exchange processes offer valuable direction for further research. These scientific questions are indeed significant and complex, making it impossible for a single paper to fully clarify

all aspects of fronts. For instance, previous studies have highlighted the disputable nature of the poleward shift of fronts in the Southern Ocean due to differing definitions of fronts (Chapman et al., 2020). Moreover, elucidating the effects of fronts on cross-shelf exchange necessitates the integration of multiple methods, including extensive voyage surveys and numerous simulations (Brink, 2016; Nencioli et al., 2016). We are grateful for acknowledging our method and digital map of persistent fronts as useful tools for investigating these questions in detail. However, this study primarily focuses on the long-term changes in the occurrence and intensity of persistent fronts from a global perspective. Therefore, while the scientific questions you raised are important, they extend beyond the scope of this work. Nevertheless, our digital map of persistent fronts provides a foundation for further studies aimed at elucidating the poleward shift, long-term or interdecadal change mechanisms, cross-shelf exchange, and other dynamic or ecological effects of each specific persistent front in the future.

Meanwhile, we also conducted some simple analyses to investigate the shift of persistent fronts, and we found that their positions have undergone slight changes in many LMEs. Fig. R1 provides a typical example in the Southwest Atlantic. This result is not surprising, as the formation of many persistent fronts in LMEs is primarily related to topographical changes (Belkin et al., 2009). We also acknowledge that some persistent fronts not confined by marine topography in the open sea may exhibit a meridional shift (Wu et al., 2018), but this study focuses on persistent fronts in LMEs. Therefore, we have added some discussion to illustrate the potential applications of our map (Lines 354-360), as follows: *“In addition to these long-term changes, persistent fronts may also exhibit seasonal and interdecadal variations in response to regional environmental changes and large-scale climate events³¹, and their positions may demonstrate spatial shifts under climate change²⁶. Our digital map of persistent fronts lays the groundwork for future studies aimed at elucidating the spatial shift, long-term or interdecadal change mechanisms, cross-shelf exchange, and other dynamic or ecological effects of each specific persistent front.”* In addition, based on your suggestion, we also analyzed the long-term changes in SST gradients and found that SST gradients also exhibit significant enhancement corresponding to stronger

persistent fronts along boundary currents and upwelling systems. It is challenging to ascertain whether increasing gradients can limit cross-shelf exchange solely based on satellite observations. We have added this discussion (Lines 228-232) as follows: *“Meanwhile, we also observed that more frequent and stronger persistent fronts along boundary currents and upwelling systems correspond to enhanced SST gradients (Supplementary Fig. 7), which may alter the cross-shelf exchange processes and exert a significant impact on the ecosystem in shelf waters³⁸.”*

Fig. R1 Preliminary comparison of identified persistent frontal pixels between 1982-1993 (blue lines) and 2012-2021 (red lines).

Belkin, I. M., Cornillon, P. C., & Sherman, K. (2009). Fronts in large marine ecosystems. *Progress in Oceanography*, 81(1-4), 223-236.

Brink, K. H. (2016). Cross-shelf exchange. *Annual review of marine science*, 8, 59-78.

Chapman, C. C., Lea, M. A., Meyer, A., Sallée, J. B., & Hindell, M. (2020). Defining

Southern Ocean fronts and their influence on biological and physical processes in a changing climate. *Nature Climate Change*, 10(3), 209-219.

Nencioli, F., Petrenko, A. A., & Doglioli, A. M. (2016). Diagnosing cross - shelf transport along an ocean front: An observational case study in the Gulf of Lion. *Journal of Geophysical Research: Oceans*, 121(10), 7218-7243.

Wu, B., Lin, X., & Qiu, B. (2018). Meridional shift of the Oyashio Extension front in the past 36 years. *Geophysical Research Letters*, 45(17), 9042-9048.

Q2: Lines 44-45: I'd also update reference #4 to something more recent (see Li et al. 2023), which states that Western Boundary Currents are unlikely to be intensifying, but changing state.

A: Thank you for your suggestion. We have incorporated this good citation (Line 47).

Q3: Line 65: I think it's worth noting that fronts can transport heat out of the ocean, as well as into the interior.

A: Thank you for your comment. We have revised it as "*have been considered as ducts responsible for transporting heat, carbon, oxygen, and other climatically important gases into the deep ocean, as well as facilitating heat transfer out of the ocean*" (Line 63-65).

Q4: Lines 135-138: Are the large numbers of persistent fronts identified in the named LMEs simply due to the fact that those LMEs cover larger areas (such as the Antarctic LME which is circum-hemispheric)? Could some measure of fronts per unit area be derived?

A: Thank you for your suggestion. We have provided the results about fronts per unit area in Supplementary Fig. 2b. We have included some descriptions as follows (Line 138-141): *When considering the coverage area of each LME, the Barents Sea LME 20 exhibited the highest number of frontal pixels per unit area, whereas the Antarctica LME 61 demonstrated a moderate level due to its expansive coverage area*

(Supplementary Fig. 2b).

Q5: Line 225: As above, the paper(s) by Junde Li and collaborators could be useful to cite here, as they've specifically studied the dynamic response of boundary currents to climate change.

A: Thank you for your suggestion. We have included this citation in the sentence (Line 234).

Q6: Lines 232-233: WBCs also tend to be highly eddying, and the interaction of eddies with the coast, the boundary currents, and other eddies, tends to lead to the formation of fronts;

A: Thank you for your suggestion. We have included this sentence as follows (Line 243-245): *Frequent eddy activity around boundary currents can interact with submarine topography, the boundary currents, and other eddies, often resulting in the formation of fronts*^{5,40}.

Reviewer #3 (Remarks on code availability):

Q7: Code requires more comments and a brief document explaining what it does and how it can be run. Variable names (for example, 'mm') are not clear, which makes the code difficult to follow.

It appears that the code requires use of the image-processing toolbox, which is not standard in Matlab, and could potentially stop an interested user from applying the code.

A: Thank you for your feedback. We have made several enhancements to our code based on your suggestions. We have added detailed comments and instructions to the code file, as well as descriptions of intermediate variables. Users can now simply input 'sstf' (frontal occurrence, %) and 'mask' (ocean mask) to run the code and obtain the results of persistent frontal pixels. For more details, please refer to <https://doi.org/10.5281/zenodo.10210578>. Regarding the functions used from the Image Processing Toolbox, especially for many morphological operations, these

functions are indispensable and challenging to replace with other standard functions. Matlab conveniently integrates the Image Processing Toolbox, and users can easily install it by accessing the 'Add-Ons' button in the main interface of Matlab.